# A Novel Strategy to Fit and Validate Physiological Models: A Case Study of a Cardiorespiratory Model for Simulation of Incremental Aerobic Exercise

**DOI:** 10.3390/diagnostics13050908

**Published:** 2023-02-27

**Authors:** Carlos A. Sarmiento, Leidy Y. Serna, Alher M. Hernández, Miguel Á. Mañanas

**Affiliations:** 1Bioinstrumentation and Clinical Engineering Research Group, Bioengineering Department, Engineering Faculty, Universidad de Antioquia UdeA, Calle 70 # 52-51, Medellin 050016, Colombia; 2Departament d’Enginyeria de Sistemes, Automàtica i Informàtica Industrial (ESAII), Universitat Politècnica de Catalunya, 08028 Barcelona, Spain; 3CIBER de Bioingeniería, Biomateriales y Nanomedicina (CIBER-BBN), 28040 Madrid, Spain

**Keywords:** fitting, mathematical modeling, sensitivity analysis, parameter estimation, cardiorespiratory system, aerobic exercise

## Abstract

Applying complex mathematical models of physiological systems is challenging due to the large number of parameters. Identifying these parameters through experimentation is difficult, and although procedures for fitting and validating models are reported, no integrated strategy exists. Additionally, the complexity of optimization is generally neglected when the number of experimental observations is restricted, obtaining multiple solutions or results without physiological justification. This work proposes a fitting and validation strategy for physiological models with many parameters under various populations, stimuli, and experimental conditions. A cardiorespiratory system model is used as a case study, and the strategy, model, computational implementation, and data analysis are described. Using optimized parameter values, model simulations are compared to those obtained using nominal values, with experimental data as a reference. Overall, a reduction in prediction error is achieved compared to that reported for model building. Furthermore, the behavior and accuracy of all the predictions in the steady state were improved. The results validate the fitted model and provide evidence of the proposed strategy’s usefulness.

## 1. Introduction

Mathematical modeling is an interdisciplinary field that, applied to medicine, has allowed a better understanding of physiological functions and relationships. Its support in this field has been related to the education and training of clinical staff, the identification, monitoring, and treatment of diseases, and equipment development [1,2,3,4,5]. Mathematical models of physiological systems constitute approximations fitted to bounded populations and specific sets of physiological events or stimuli, so their practical and conceptual utility depends on their ability to predict experimental measurements of physiological variables [2,3,6]. Its mathematical complexity increases with accuracy and physiological relevance, so the number of parameters and equations is considerably high for describing the regulation mechanisms of the most complex physiological systems [7,8].

The cardiorespiratory system stands out as one of the most relevant physiological systems regarding the diagnosis, treatment, monitoring, and prevention of diseases, mainly due to the importance and usefulness of their related variables for the identification of the correct state and functioning of the organism [9,10,11]. The cardiorespiratory variables result from different and complex regulatory mechanisms focused on maintaining correct physiological functioning, even under different external conditions, such as diseases [12,13]. Different mathematical models have focused on the cardiorespiratory system, highlighting its support for personalized diagnosis applications due to its ability to be fitted to a subject or population under a specific stimulus [1,8,14].

Physical exercise is a natural stimulus that generates significant cardiorespiratory variations that cannot be evidenced under resting conditions and constitute a well-defined characteristic pattern in healthy human subjects [15,16]. Several mathematical models of the cardiorespiratory system have been proposed, but few of them can correctly represent the human response to exercise. The handicaps of those reported models to simulate the cardiorespiratory response to this stimulus are diverse. For instance, although some physiological models [8,17] detail some behaviors, they do not consider important mechanisms of neuronal regulation and omit the prediction of essential variables related to the exercise response. In contrast, other more complete models [18,19], despite including a complete description of the systems and allowing the prediction of variables useful for exercise analysis, are not adequate for predicting the human response under the stimulus due to the lack of related physiological mechanisms. For this aspect, models of the respiratory [20,21] and cardiovascular [15] controller, and even more recent works such as the model by [22], involve mechanisms and dynamics related to the response under aerobic exercise. However, the application of these models in a complex study is limited by their specificity and specialization in only one of the systems involved.

Although mathematical models can describe the physiological response under a specific stimulus, their computational simulation using parameters’ nominal values constrains their use in populations with the same characteristics and conditions used during their development [14,23]. Fitting processes allow the personalization of physiological models to a specific subject or population by estimating the parameter values that minimize the difference between predictions and experimental observations [6,8]. Applying these techniques to physiological models could involve high complexity due to the large number of parameters that need to be identified. In addition, the restricted number of experimental observations could imply multiple solutions, and several parameter values could not have a physiological sense [8].

Different techniques to solve the fitting problem of models with many parameters have been proposed. They all focused on identifying the minimum number of parameters that must be fitted to correctly predict the experimental response. In [6], a deterministic sensitivity analysis focused on those parameters that are resolvable in the presence of noise, based on the effect of parameter value variations on the interest variables, was described. In a different approach, Ref. [24] validated a subset selection method that focused on finding the well-conditioned independent parameters for reliable identification using experimental measurements. Refs. [6,24] do not consider the evaluation of different levels of stimulus. In contrast, Ref. [8] proposed a classification technique based on the overall sensitivity of the model regarding parameter variations and considering different stimulus levels [23]. However, this proposal underestimates sensitivity measures by considering the sum of changes in different directions and assuming that all variables contribute equally to the error. Although the subsequent fitting could be designed for a steady state, its sensitivity computation includes time lags that are not consistent.

The validation procedure aims to verify whether the model predictions agree with the experimental data for the defined conditions and disturbances. For this purpose, measures that consider the transient and steady-state response features of the analyzed physiological variables are usually implemented. Although there is no consensus on the proper methodology for validating mathematical models of physiological systems, different works coincide with the criteria proposed by [25]. Such criteria allow validating a model when the dynamic of its responses is consistent with the expected behavior and the values in the steady state are accurate.

This work aims to propose a fitting and validation strategy for physiological models with many parameters that consider different populations and stimuli. This strategy includes several techniques and approaches to classify, select, and optimize model parameters under steady-state conditions. A model validation methodology is also presented. Some described methods are new proposals or correspond to improvements to previously reported methods. The strategy application in a cardiorespiratory model during dynamic aerobic exercise is presented and described as a case study.

This paper is structured as follows. First, the fitting and validation strategies are presented. It describes in detail the procedures for the classification, selection, and optimization of parameters and methodology validation. Second, a case study is presented. It includes a qualitative description of the cardiorespiratory model and the simulation, fitting, and validation. Third, predictions at different levels of aerobic exercise, both with the fitted model and the nominal values, are compared with experimental data obtained from healthy subjects under a cardiopulmonary exercise test [26]. Finally, the proposed strategy and the model performance results are discussed.

## 2. Materials and Methods

### 2.1. Fitting and Validation Strategy

The strategy proposed in this work consists of applying three main procedures or steps that must be done sequentially: (a) classification and selection of parameters, (b) model fitting, and (c) model validation. The first procedure involves classifying the model parameters according to some predefined roles in the model (gain, threshold, initial conditions, etc.) and selecting the most relevant ones according to different model-fitting approaches. The second one consists of four sequential stages of parameter value identification related to the model fitting regarding the available experimental information, the overall accuracy, the specific prediction of each variable, and the predictions’ behavior concerning the evaluated stimulus. The final procedure focuses on validating the fitted model regarding its predictions’ accuracy, behavior, and transient and steady-state regimes. Each procedure is presented in Section 2.1.1, Section 2.1.2 and Section 2.1.3, and its application is described according to the case study. Figure 1 shows a schematic summary of the strategy, in which the main procedures and their respective stages are shown.

#### 2.1.1. Classification and Selection of Parameters

This procedure aims to highlight and select the parameters that can be considered best-fit candidates. The selection of parameters is justified according to their relationship to available experimental data, role in the model, identifiability, sensitivity to variations, and their relationship to the stimulus evaluated. Initially, it reduces the number of parameters depending on their role in the model and the objective of fitting predictions in steady-state conditions. Subsequently, the resulting parameters are classified and selected according to two reported techniques and established criteria regarding the model fitting approaches.


**Classification and Selection of Parameters by Role**


It consists of an initial classification of the model parameters according to their roles. It comprises five parameter classes generally found in structured models of physiological systems. They correspond to (i) time constants, i.e., parameters related to transient response; (ii) conversion parameters, which are constant values related to equivalences among measurement units; (iii) covariates, corresponding to values that allow defining simulation conditions regarding external disturbances, environment conditions, and features of the population to be simulated; (iv) initial values, corresponding to initial conditions of the model variables, usually required as initial states of integrators and whose action mainly affect the temporal characteristics of the responses before reaching the model steady state; and (v) gain and thresholds, that either module or saturate variables related to model mechanisms, i.e., the weighting of the chemoreceptors response to set the parasympathetic and sympathetic activity regarding the regulation of peripheral resistances in cardiovascular control models.

The selection of parameters comprises choosing those parameters that mainly define the steady-state behavior of the cardiorespiratory system response and, therefore, can be used to fit the model response in such a condition. The time constant parameters and initial values are discarded, considering that the proposed fitting procedure focuses on the response once the steady state is reached. Conversion parameters do not need fitting because of their nature and meaning. Therefore, only gain and threshold parameters, whose variations have physiological sense, and are correlated with experimental conditions, and covariates will be considered for the subsequent selection procedures and fitting process. Covariates are only used for the standardization of the simulation conditions.


**Parameter Classification Techniques**


The proposed strategy involves modifying and implementing two of the most widely implemented parameter classification techniques for fitting physiological models [8,23]. A detailed description of each is presented below.


*Subset Selection of Parameters*


It is a technique based on the classification of model parameters according to how well-conditioned or ill-conditioned they can be identified [27]. Well-conditioned parameters correspond to those that can be reliably estimated from the constrained experimental data, while ill-conditioned ones are those for which there are multiple fitting solutions.

In this study, the subset selection of parameters is based on QR Factorization with the Column Pivoting method [28]. It was selected because it has been widely implemented in different physiological models [2,8,23,29], presented one of the best results regarding cardiorespiratory models among the other methods reported [28], and considers the difference between model predictions and experimental data as a reference [2]. This technique is based on the solution of optimization problems using gradient techniques. It establishes a ranking of the well-conditioned parameters for identification by analyzing the interdependencies in the Jacobian.

This technique is complemented by integrating the different stimulus exercise levels (according to the experimental data) and variations in the parameters previously selected in the appropriate physiological range (around their nominal values). These additions would allow the selection of parameters that are more consistent with the desired fit.

According to the above, the Jacobian is calculated according to Equation (1).
(1)$$r_{\mathrm{kl}}^{\prime }\left(u_{\mathrm{jl}}\right)=\frac{\partial r_{\mathrm{ikl}}\left(u_{\mathrm{jl}}\right)}{\partial u_{\mathrm{jl}}},$$
where the indexes i, j, and k represent the variable, parameter, and stimulus level to be evaluated, respectively; l is an index that identifies the parameter variation regarding its nominal value. $r_{\mathrm{kl}}^{\prime }$ is the Jacobian for a specific stimulus level and parameter value variation, formed as the matrix of variations of the differences for each variable $r_{\mathrm{ikl}}$ regarding the change of each parameter for a specific variation level $u_{\mathrm{jl}}$; r corresponds to the difference between each experimental data and the model prediction given each parameter’s change for a particular level of variation. The singular value decomposition is calculated for $r_{\mathrm{kl}}^{\prime }$, according to Equation (2).
(2)$$r_{\mathrm{kl}}^{\prime }\left(u_{\mathrm{jl}}\right)=U_{\mathrm{jkl}}\Sigma_{\mathrm{jkl}}V_{\mathrm{jkl}}^{T},$$
where $U_{\mathrm{jkl}}$ is the matrix of left singular vectors, $\Sigma_{\mathrm{jkl}}$ is the diagonal matrix of singular values of $r_{\mathrm{kl}}^{\prime }\left(u_{\mathrm{jl}}\right)$ in decreasing order, $V_{\mathrm{jkl}}$ is the matrix of the right singular vectors, and T denotes the matrix transpose. Matrix $V_{\mathrm{jkl}}$ must be partitioned, as expressed in Equation (3).
(3)$$V_{\mathrm{jkl}}=\left[\begin{array}{cc} V_{\mathrm{jkl},\rho \left(j,k,l\right)} & V_{\mathrm{jkl},W-\rho \left(j,k,l\right)} \end{array}\right],$$
where W is the total number of parameters analyzed, and ρ(j,k,l) is a numerical rank that indicates the number of maximally independent columns of $r_{\mathrm{kl}}^{\prime }\left(u_{\mathrm{jl}}\right)$. ρ(j,k,l) is equivalent to the number of parameters that can be identified given the model output and can be determined by the selection of the smallest allowed singular value according to the relation expressed in Equation (4) [23].
(4)$$\frac{\sigma_{\mathrm{jkl},\rho \left(j,k,l\right)}}{\sigma_{\mathrm{jkl},1}}={\sigma N}_{\mathrm{jkl},\rho \left(j,k,l\right)}>\varepsilon ,$$
where $\sigma_{\mathrm{jkl},\rho \left(j,k,l\right)}$ is the singular value for ρ(j,k,l), $\sigma_{\mathrm{jkl},1}$ is the largest singular value, ${\sigma N}_{\mathrm{jkl},\rho \left(j,k,l\right)}$ is the normalized singular value for ρ(j,k,l), and ε is a tolerance value that allows differentiating the most significant eigenvalues.

The parameters associated with the ρ(j,k,l) highest singular values are found using QR decomposition with column pivoting, according to Equation (5).
(5)$$V_{\mathrm{jkl},\rho \left(j,k,l\right)}^{T}P_{\mathrm{jkl}}=Q_{\mathrm{jkl}}R_{\mathrm{jkl}},$$
where $P_{\mathrm{jkl}}$ is a permutation matrix, $Q_{\mathrm{jkl}}$ is an orthogonal matrix and the first ρ(j,k,l) columns of $R_{\mathrm{jkl}}$ form an upper triangular matrix with diagonal elements in decreasing order. $P_{\mathrm{jkl}}$ is then used to reorder the parameters according to Equation (6).
(6)$${\hat{u}}_{\mathrm{jkl}}=P_{\mathrm{jkl}}^{T}u_{I},$$
where $u_{I}$ is an identification vector of the parameters and ${\hat{u}}_{\mathrm{jkl}}$ is the vector of the parameters reordered, which is partitioned regarding ρ(j,k,l), as presented in Equation (7).
(7)$${\hat{u}}_{\mathrm{jkl}}=\left[\begin{array}{cc} {\hat{u}}_{\mathrm{jkl},\rho \left(j,k,l\right)} & {\hat{u}}_{\mathrm{jkl},W-\rho \left(j,k,l\right)} \end{array}\right],$$
where ${\hat{u}}_{\mathrm{jkl},\rho \left(\mathrm{jk},l\right)}$ is a vector containing the ρ(j,k,l) estimable parameters and ${\hat{u}}_{\mathrm{jkl},W-\rho \left(k,l\right)}$ are the parameters that would be fixed at nominal values.

The standard 2-norm of the normalized singular value for each parameter in each ranking obtained for each parameter variation must be determined to obtain a general ranking independent of the stimulus level, as presented in Equation (8).
(8)$$Z_{\mathrm{jl}}={\left\|Z_{\mathrm{jlk}}\right\|}_{2}=\sqrt{\frac{1}{K}\sum_{k=1}^{K}{\left(Z_{\mathrm{jkl}}\right)}^{2}},$$
where $Z_{\mathrm{jl}}$ corresponds to the standard 2-norm of the normalized singular value of the parameter j for the variation level l, $Z_{\mathrm{jkl}}$ is the normalized singular value of the parameter j for the stimulus level k and the parameter variation level l, and K is the number of stimulus levels.

Finally, the result of this technique corresponds to the presented in Equation (9).
(9)$$Z_{j}={\left\|Z_{\mathrm{jl}}\right\|}_{2}=\sqrt{\frac{1}{L}\sum_{l=1}^{L}{\left(Z_{\mathrm{jl}}\right)}^{2}},$$
where $Z_{j}$ is the standard 2-norm of $Z_{\mathrm{jl}}$ of the parameter j, and L is the total number of parameter variations.

This technique is usually implemented with sensitivity analysis techniques because their approaches complement each other regarding the problem solution. QR decomposition identifies the parameters to which the model is sensitive as a group, while sensitivity analysis finds the parameters to which the model is individually sensitive [2].


*Sensitivity Analysis*


Sensitivity analysis involves the integration and improvement of the techniques introduced by [6,8,23]. It comprises a deterministic analysis that evaluates the global model and variables’ sensitivity to variations in model parameters and stimulus levels. It also involves an experimental data dependency term that weighs such sensitivity by the error reached by the model at each parameter variation and stimulus level.

The model variable’s sensitivity regarding each parameter variation is based on the standard local differential equation described by [30] and used by [8,19,23,31,32] to calculate time-dependent sensitivities in cardiovascular and respiratory models. Equation (10) shows the computation of this sensitivity.
(10)$${s_{\mathrm{ij}}\left(t,u\right)|}_{u=u_{n}}=\frac{\partial Y_{i}\left(t,u_{j}\right)}{\partial u_{j}}{\frac{u_{j}}{Y_{i}\left(t,u\right)}|}_{u=u_{n}};u_{j},Y_{i}\left(t,u\right)\neq 0\;,$$
where $s_{\mathrm{ij}}$ represents the relative sensitivity of the variable $Y_{i}$ to parameter $u_{j}$, which is dimensionless by the ratio between the parameter $u_{j}$ and the variable $Y_{i}$ values at nominal conditions (no parameter variation). n refers to the parameter’s nominal value.

This work proposes modifying the measurement of relative sensitivity to make it independent of the time and dependent on the stimulus. Therefore, an approach based on steady-state conditions at different stimulus levels is used. Relative sensitivity is evaluated by varying each parameter over a range around its nominal value, while the others are kept at their nominal values. Equation (11) shows the proposed relative sensitivity measure. Time independence avoids significant differences between sensitivity measures due to time lags, considering that the fitting model procedure is focused on minimizing the steady-state differences between the experimental measurements and the model predictions. The evaluation of the stimulus is included by considering the influence of its variations on sensitivity measures.
(11)$$s_{\mathrm{ijk}}=\left|\frac{\partial Y_{\mathrm{ik}}\left(u_{j}\right)}{\partial \left(u_{j}\right)}\right|\left|\frac{u_{\mathrm{jn}}}{Y_{\mathrm{ik}\;}(u_{n})}\right|,$$
where, i, j, and k are indexes that refer to the analyzed variable, parameter, and stimulus level. $s_{\mathrm{ijk}}$ is a scalar value of the relative sensitivity; Y is the variable value in the steady state; u is the parameter value; n refers to the parameter’s nominal value. Therefore, Equation (11) evaluates changes in the variable $Y_{i}$ regarding variations in the parameter $u_{j}$ at the stimulus level k (first quotient), which is dimensionless by the ratio between the parameter’s nominal values $u_{\mathrm{jn}}$ and the variable $Y_{i\;}$ at the stimulus level k (second quotient).

Deriving sensitivity equations can be tedious and error-prone for large systems, mainly when they involve nonlinear features, such as those analyzed in this case study. Alternatively, Equation (11) can be solved using a computational approach consisting of a simple finite difference method. A numerical approximation of the derivatives, which also considers the variation of the parameter relative to its nominal value, is expressed in Equation (12).
(12)$$s_{\mathrm{ijlk}}\approx \left|\frac{Y_{\mathrm{ik}}\left(u_{\mathrm{jn}}+h_{l}u_{\mathrm{jn}}\right)-Y_{\mathrm{ik}}\left(u_{n}\right)}{\left(u_{\mathrm{jn}}+h_{l}u_{\mathrm{jn}}\right)-\left(u_{\mathrm{jn}}\right)}\right|\left|\frac{u_{\mathrm{jn}}}{Y_{\mathrm{ik}\;}(u_{n})}\right|,$$
where $s_{\mathrm{ijkl}}$ is a scalar value of the sensitivity for the variable $Y_{i}$ regarding the variation $h_{l}$ in the parameter $u_{j}$ at the stimulus level k. h is the vector of change proportions of the parameter regarding its nominal value. This expression can be reduced and organized as expressed in Equation (13).
(13)$$s_{\mathrm{ijlk}}\approx \left|\frac{Y_{\mathrm{ik}}\left(\left(h_{l}+1\right)u_{\mathrm{jn}}\right)-Y_{\mathrm{ik}}\left(u_{n}\right)}{Y_{\mathrm{ik}\;}(u_{n})}\right|\left|\frac{1}{h_{l}}\right|={\mathrm{DY}}_{\mathrm{ijlk}}\left|\frac{1}{h_{l}}\right|,$$
where the first quotient of the equation, also identified as ${\mathrm{DY}}_{\mathrm{ijkl}}$, is a dimensionless term representing the rate of change of the variable $Y_{i}$ when a percentual variation $h_{l}$ in the parameter $u_{j}$ is applied at the stimulus level k. The second quotient is a dimensionless term representing a weighting factor, giving heavier importance to slight parameter variations. Therefore, the sensitivity $s_{\mathrm{ijlk}}$ corresponds to a dimensionless scalar value that measures the weighted relative variations of the analyzed variables according to four degrees of freedom.

Standard 2-norm is applied to $s_{\mathrm{ijlk}}$ to obtain a measure of the relative sensitivity independent of the parameter variations, as expressed in Equation (14). As a result, a positive dimensionless scalar value related to the relative sensitivity’s mean trend is obtained for all parameters.
(14)$$s_{\mathrm{ijk}}={\left\|s_{\mathrm{ijlk}}\right\|}_{2}=\sqrt{\frac{1}{L}\sum_{l=1}^{L}{\left({\mathrm{DY}}_{\mathrm{ijlk}}\left|\frac{1}{h_{l}}\right|\right)}^{2}},$$
where L is the total number of parameter variations or elements of the vector h.

The stimulus level independency relative sensitivity is calculated according to Equation (15).
(15)$$s_{\mathrm{ij}}={\left\|\frac{s_{P,\mathrm{ijk}}}{\max\left(s_{\mathrm{ik}}\right)}\right\|}_{2}=\sqrt{\frac{1}{K}\sum_{k=1}^{K}{\left(\frac{P_{\mathrm{ik}}\cdot s_{\mathrm{ijk}}}{\max\left(s_{\mathrm{ik}}\right)}\right)}^{2}},$$
where K is the total number of stimulus levels; $\max\left(s_{\mathrm{ik}}\right)$ is the maximum relative sensitivity obtaining for the variable i at the stimulus level k considering all parameters; $s_{P,\mathrm{ijk}}$ is the weighted relative sensitivity; and $P_{\mathrm{ik}}$ is a weighting factor representing the variable error at the stimulus level k.

$S_{\mathrm{ij}}$ corresponds to the standard 2-norm of $s_{\mathrm{ijk}}$ normalized by its maximum value at each stimulus level k and weighted by $P_{\mathrm{ik}}$. Normalization of $s_{\mathrm{ijk}}$ allows comparing the sensitivity measures between different stimulus levels, whereas the inclusion of the weighting factor $P_{\mathrm{ik}}$ prioritizes the sensitivity obtained at the stimulus levels for which the variable’s error is more significant (i.e., errors obtained from predictions resulting from using parameter’s nominal values). As a result, a ranking of the model parameters concerning their variation impact for each variable is obtained.

$P_{\mathrm{ik}}$ is calculated according to Equation (16).
(16)$$P_{\mathrm{ik}}=\frac{E_{\mathrm{ik}}}{K\cdot E_{\mathrm{iT}}},$$
where $E_{\mathrm{ik}}$ is the error of the variable $Y_{i}$ at the stimulus level k, and $E_{\mathrm{iT}}$ is the total error for the variable $Y_{i}$ (considering all stimulus levels). Equations (17) and (18) are proposed to calculate the mentioned errors.
(17)$$E_{\mathrm{ik}}=\left|\frac{Y_{\exp,\mathrm{ik}}-Y_{\mathrm{ik}}\left(u_{n}\right)}{Y_{\exp,\mathrm{ik}}}\right|,$$
(18)$$E_{\mathrm{iT}}=\frac{1}{K}\overset{K}{\underset{k=1}{\sum }}E_{\mathrm{ik}},$$
where $Y_{\exp,\mathrm{ik}}$ represent the experimental value of the variable $Y_{i}$ at the stimulus level k.

The model’s total sensitivity to each parameter is calculated according to Equation (19).
(19)$$s_{j}={\left\|\frac{s_{P,\mathrm{ij}}}{\max\left(s_{i}\right)}\right\|}_{2}=\sqrt{\frac{1}{I}\sum_{i=1}^{I}{\left(\frac{P_{i}\cdot s_{\mathrm{ij}}}{\max\left(s_{i}\right)}\right)}^{2}},$$
where I is the total number of variables evaluated; $\max\left(s_{i}\right)$ is the maximum value of sensitivity among all the parameters for the variable $Y_{i}$; $s_{P,\mathrm{ij}}$ is the weighted sensitivity of each parameter $u_{j}$ for the variable $Y_{i}$; $P_{i}$ is the weighting factor related to the variable and it is calculated according to the Equation (20).
(20)$$P_{i}=\frac{E_{i}}{I\cdot E_{T}},$$
where $E_{i}$ is the error of the variable i, and $E_{T}$ is the total model error. Equations (21) and (22) are proposed to calculate the mentioned errors.
(21)$$E_{i}=\sqrt{\frac{1}{K}\sum_{k=1}^{K}{\left(\frac{Y_{\exp,\mathrm{ik}}-Y_{\mathrm{ik}}\left(u_{n}\right)}{Y_{\exp,\mathrm{ik}}}\right)}^{2}},$$
(22)$$E_{T}=\frac{1}{I}\overset{K}{\underset{k=1}{\sum }}E_{i},$$

$S_{j}$ corresponds to the standard 2-norm of $S_{\mathrm{ij}}$ normalized by the maximum sensitivity obtained in each variable and weighted by $P_{i}$. Normalization of $s_{\mathrm{ij}}$ allows comparing the sensitivity measures of the parameters among different variables, whereas $P_{i}$ prioritizes the sensitivities of the parameters for which the variable’s error is more significant. As a result, a ranking of the model parameters representing the effect of their variation on the model’s whole output is obtained.


**Parameter Selection Criteria**


This procedure focuses on applying the selection criteria of the classified parameters concerning four different fitting approaches.


*Selection for the Standardization of Simulation Conditions*


It consists of selecting the model’s parameters, which can be determined from the available experimental data. It involves those related to the subjects’ characteristics, the stimulus evaluated, or the environmental conditions. They can be established by direct equivalence or by applying previously validated equations.


*Selection for the Base Fitting Approach*


It comprises selecting the set of parameters for which the model has the highest global sensitivity. This selection aims to reduce the model’s total prediction error by considering the parameter’s overall effect on model behavior. They correspond to the union of the set of parameters obtained in subset selection (i.e., those parameters that have been identified as well-conditioned to be adjusted, Equation (9)) and total sensitivity techniques (i.e., those parameters whose variations generate significant changes in the model output, Equation (19)). In this work, this selection is defined according to the following criteria.

Parameters from the subset selection ranking are chosen according to the criterion defined in Equation (4), considering ε as the square root of the termination tolerance on the function evaluation defined for the fitting procedure (Table 1).Parameters from the total sensitivity ranking are chosen in descending order until at least one is obtained for each system and controller of the model, including those already selected in the previous step.


*Selection for a Specific Fitting Approach*


It corresponds to the set of parameters for each variable’s highest sensitivity at the individual level. Its optimization aims to modify each variable’s predictability to bring it closer to the respective experimental data without significantly affecting the other predictions. It is based on relative sensitivity measures independent of the stimulus level (Equation (15)). Based on the following criteria, only one parameter is selected by each ranking obtained for the evaluated physiological variables.

Remove the parameters selected for the base fitting approach from each variable ranking.Remove the parameters of systems and controllers that are not directly related to regulating the variable of interest.Select only those parameters whose sensitivity is high for the variable of interest and low for the remaining ones. Parameters with high sensitivity for other variables could be selected if those variables belong to or are dependent on the same system or controller of the variable of interest.


*Selection for the Stimulus-Related Fitting Approach*


This corresponds to the parameter selection criteria that relate the stimulus to the regulation mechanisms addressed in the model under study. Their selection is based on the parameters’ role concerning the mechanisms mainly associated with the stimulus, highlighting the weighting factors or gains that link them to regulatory activities.

Eight mechanisms directly related to the exercise stimulus were evaluated in this work. Only one parameter per mechanism was selected.

#### 2.1.2. Model Fitting

Fitting a parametric model involves solving an optimization problem in which the values of the set of parameters minimizing the differences between experimental data and model predictions are identified [6]. The identification of the parameter values in this strategy results in applying an optimization algorithm in three stages that must be carried out sequentially: first, a base optimization; second, a specific optimization; and third, a stimulus-related optimization. Each stage focuses on the value identification of a specific and reduced number of parameters. It is proposed to apply the following procedures before the mentioned optimization stages to obtain correct, fast, and physiological meaning results: the standardization of simulation conditions, the selection and parameterization of the optimization algorithm, the definition of the parameter evaluation ranges, and the choice of a metric to evaluate the model’s goodness of fit [33]. Each procedure and optimization stage is detailed below.


**Standardization of Simulation Conditions**


Its objective is to bring the simulation conditions close to the data used as a fitting reference. To do this, it proposes modifying some model parameters using the values obtained from the experimental data. It replaces the selected parameters’ values with the direct equivalences or results of applying previously validated equations. Its implementation has been previously presented in works on the design and fitting of physiological models, e.g., for the estimation of blood volumes and mechanical characteristics of cardiovascular and respiratory systems [8,34,35]. It reduces the number of fitting candidates and, therefore, the complexity of the optimization problem and the possibility of multiple solutions.


**Optimization Algorithm**


Different optimization algorithms reported could be applied for the fitting of physiological models. They are divided into those that use deterministic or stochastic methods. Their correct selection depends on the model’s specific characteristics under study and the validation requirements [33,36].

In this paper, an evolutionary strategy with a covariance matrix and adaptation (CMA-ES) was selected. It is a stochastic global optimization algorithm based on adaptive and evolutionary strategies [37]. This algorithm was selected considering the best convergence speed, precision, and accuracy results reported in a comparative study concerning a physiological model similar to the case study [33].


**Parameterization of the Optimization Algorithm**


This corresponds to the parameter assignment of the selected optimization algorithm. It is generally related to the number of evaluations, iterations, and error tolerances. These parameters are specific to the algorithm and must be defined based on the fitting’s strictness and computational cost.

The parameter values used for implementing the CMA-ES algorithm are presented in Table 1. These values are similar to those reported by [33], considering the similarity regarding the case study model and that the fitting’s strictness is analogous to the desired one.


**Evaluation Ranges of the Parameters**


The optimization algorithms seek to identify model parameters through strategic variations of their values. The variation ranges should depend on the parameter’s specific function concerning the associated model mechanism since not all possible values provide a consistent result. For physiological models, in addition to obtaining predictions close to experimental data, it is desirable to find optimized values that reflect consistent physiological conditions or characteristics, further constraining the optimization problem [33].

According to the above, this work proposes the following criteria to determine the evaluation range for each selected parameter:Determine an initial variation range for each parameter in proportion to its nominal value. The range bounds depend on the expected closeness between the optimized and nominal values. Such closeness can be estimated overall by considering the parameter’s relationship with the condition for which its nominal value was defined and the nature of the experimental dataset for which it is intended to optimize, e.g., variation of respiratory mechanics parameters during rest and exercise [38]. The variation range is recommended to contain the values evaluated in the parameter selection techniques.Evaluate the values’ constraints from the equations describing the model mechanisms associated with each parameter and redefine the previously established bounds.Redefine the variation range’s bounds if the reported information relates the parameter to the experimental conditions. Since not all model parameters are directly related to physiological measures, i.e., those from empirical equations that have been fitted to experimental data, evaluating their variations considering different works in which it has been used is necessary.


**Performance Evaluation**


It uses a metric that compares the model’s predictions with the experimental data. The optimization algorithm uses this measure as a criterion function to identify the goodness of fit of the model predictions for each parameter’s optimization. Different metric options have been applied in several works related to physiological models. However, the most commonly used is based on the root mean square error (RMSE) [6,12,33], and its application is justified because it is considered more suitable for revealing differences in model performance [6,39].

This paper modified the RMSE metric to consider all analyzed variables at different stimulus levels, as expressed in Equation (23).
(23)$$\mathrm{CF}=\frac{1}{I}\sum_{i=1}^{I}\sqrt{\frac{1}{K}\sum_{k=1}^{K}{\left(\frac{y_{\exp_{i,k}}-y_{{\mathrm{sim}}_{i,k}}}{y_{\exp_{i,k}}}\right)}^{2}}\;,$$
where CF denotes the cost function for the optimization algorithm; $y_{\exp}$ and $y_{\mathrm{sim}}$ represent the variable’s experimental and simulated values; I and K indicate the number of variables and stimuli levels; the subscripts i and k denote each variable and stimulus level.

The error between predictions and experimental data is initially computed by obtaining a dimensionless measurement of each variable’s difference at each stimulus level. Subsequently, the standard 2-norm is calculated regarding stimulus levels to measure each variable’s mean trend. Finally, the global error is calculated as the mean value of every variable’s errors.


**Fitting Stages**


This work proposes fitting the model using three approaches. They involve those presented in the parameter selection procedure’s subsections and represent stages sequentially applied to obtain a complete fit of the case study model. In one stage, the previously fitted parameters remained fixed, and only those selected for the current one are optimized.

The first stage corresponds to a base fitting approach. It optimizes the parameters with the highest total sensitivity, whose variations have the most significant overall impact on the controllers and system outputs.

The second stage corresponds to a specific fitting approach. Each variable comprises the parameters’ optimization with the highest relative sensitivity and the least side effect for the remaining variables. This is focused on minimizing the prediction error of each variable individually.

The third stage involves a model mechanisms’ fitting approach. It optimizes the mechanisms’ parameters, mainly associated with the evaluated stimulus, to improve the predictions’ behavior.

#### 2.1.3. Validation Methodology

This procedure assesses the model predictions under conditions and disturbances that match the experimental data. Although there is no consensus on a defined validation methodology for mathematical models of physiological systems, most of the evaluations described in related works generally coincide with the criteria proposed in [25]. Therefore, model simulation results are evaluated using directional, accurate, and consistent approaches.

Different metrics can be used to evaluate the performance of a model in steady-state conditions, highlighting mean absolute error (MAE) and RMSE as the most common ones. Their selection must be related to the critical interpretation of the results and the statistical distribution of errors. However, there is no consensus on which of these is most appropriate [39,40]. In this work, the measure presented in Equation (24) was used to calculate the model’s prediction error of PE.
(24)$$\mathrm{PE}=\frac{1}{N}\sum_{v=1}^{N}\left({\mathrm{Median}}_{s,l}\left(\left|\frac{y_{\exp_{s,v,l}}-y_{{\mathrm{sim}}_{s,v,l}}}{y_{\exp_{s,v,l}}}\right|\right)\right)\times 100\%,$$
where $y_{\exp}$ and $y_{\mathrm{sim}}$ represent the experimental and predicted values; N indicates the number of variables; and s, v, and l are used to distinguish the subject, variable, and stimulus level.

The PE measurement was selected because (a) it corresponds to the PE measure used in the structural evaluation work of the case study model [26], (b) it allows straightforward interpretation of differences as proportions of experimental data, and (c) it considers median values due to the non-normal distribution of the experimental data.

Regarding dynamic transitions, changes in magnitude and response speed are usually implemented to evaluate predictions under stimulus changes [15,17,18]. Due to the above, and considering the measures adopted in previous evaluation and characterization works under the stimulus of exercise [26,41,42,43], this study considers evaluating each variable’s percentage change regarding its rest conditions and the settling time of the obtained predictions.

### 2.2. Case Study

#### 2.2.1. Cardiorespiratory Model

The cardiorespiratory model follows a multi-compartmental structure comprised of subsystems for cardiovascular circulation, respiratory mechanics and gas exchange processes, and controllers that allow cardiac and respiratory self-regulation. It results from adapting different published models to achieve a model that can correctly mimic the main cardiorespiratory variables of a healthy adult under incremental aerobic exercise. Its development, nominal parameter values, and complete description of mechanisms are not reported here for brevity. The reader can refer to the previous paper for full details [26].

The cardiovascular system includes systemic circulation, pulmonary circulation, and the heart. Systemic circulation is divided into large arteries, peripheral and venous vessels, and the vena cava. Pulmonary circulation involves the pulmonary arteries, peripheral vessels, and veins. The heart differentiates elements of the left and right compartments, where the atriums are modeled as passive compliance and the ventricles’ activity as a variable-elastance model. Changes in the pressures, resistances, blood flows, and blood volumes of the different vascular beds are regulated in response to cardiovascular regulation and exercise stimulus.

The cardiovascular control response is initially mediated by the activity of afferent pathways (baroreceptors, lung stretch receptors, and peripheral chemoreceptors), the blood flow local control, the central nervous system to acute ischemic conditions, the central respiratory neuromuscular drive, and the mechanism of the central command (evaluated as the metabolic regulation response). The efferent pathways modulate their actions as sympathetic and parasympathetic activities. Subsequently, these results, together with the central command response (I), allow the effector mechanisms to regulate heart rate (HR), peripheral resistances ($R_{\mathrm{jp}}$), unstressed venous volumes ($V_{u,\mathrm{jv}}$), and maximum end-systolic elastance of the ventricles ($E_{\max,\mathrm{jv}}$).

The respiratory system comprises respiratory mechanics, gas exchange, and a respiratory controller. Respiratory mechanics are divided into upper airway mechanics and pulmonary mechanics. Signals such as respiratory muscle pressure ($P_{\mathrm{musc}}\left(t\right)$), pleural pressure ($P_{\mathrm{pl}}$), alveolar pressure, ventilatory flow ($\dot{V}$) and tidal volume (VT) are generated in the pulmonary mechanics’ compartment according to variables of the upper airway mechanics and the parameters regulated by the respiratory controller (Nd). The upper airway compartment describes the dynamics of air from the mouth to the lungs. It allows calculating the conductance of the upper airway ($G_{aw}$), necessary for determining $\dot{V}$ and VT according to the air movement equation.

The gas exchange system describes the exchange, mixing, and transport of $O_{2}$ and ${\mathrm{CO}}_{2}$. The gas exchange and mixing compartment described the alveolar gas partial pressures (${\mathrm{PACO}}_{2}$ and ${\mathrm{PAO}}_{2}$), the arterial gas partial pressures (${\mathrm{PaCO}}_{2}$ and ${\mathrm{PaO}}_{2}$) and the gas concentrations in the arterial blood (${\mathrm{CaCO}}_{2}$ and ${\mathrm{CaO}}_{2}$). They are determined by the blood flow signals, blood flow of each peripheral compartment ($Q_{\mathrm{jp}}$) and blood flow from the pulmonary peripheral compartment ($Q_{\mathrm{pp}}$), respiratory mechanics waveforms ($\dot{V}$ and VT) and environmental conditions such as the atmospheric pressure ($P_{atm}$) and inspired fractions of dry gas ($FiCO_{2}$ and $FiO_{2}$). The gas transport compartment describes the gas concentrations in the venous blood (${\mathrm{CvCO}}_{2}$ and ${\mathrm{CvO}}_{2}$), the venous gas partial pressures (${\mathrm{PvCO}}_{2}$ and ${\mathrm{PvO}}_{2}$), the brain gas partial pressure (${\mathrm{PbCO}}_{2}$) and the tissues gas metabolic ratios (${\mathrm{MRTCO}}_{2}$ and ${\mathrm{MRTO}}_{2}$). It also involves a metabolically related neural drive component to ventilation (MRV), from which information related to metabolic changes during exercise for respiratory regulation processes is provided.

The respiratory controller performs regulation processes based on the chemical and mechanical optimization approaches described by a ventilatory controller and a respiratory pattern optimizer. The ventilatory controller estimates the ventilatory demand at the end of each respiratory cycle in response to the average activity of the central and peripheral chemoreceptors for CO_2_ and O_2_, the neural drive ventilation related to metabolism (MRV) and the basal alveolar ventilation. The respiratory pattern optimizer estimates the variables related to the global breathing pattern, specifically the breathing frequency (BF) and the inspiratory time (TI), and the parameters defining the respiratory mechanics’ waveforms according to the ventilatory demand and the minimization of the mechanical work of breathing.

The model has different parameters that can be configured as inputs to simulate a healthy human subject’s response with specific characteristics under several stimuli, e.g., at different levels of aerobic exercise and environmental conditions. Carbon dioxide output ($\dot{V}CO_{2}$) and oxygen uptake ($\dot{V}O_{2}$) can be used to represent different aerobic exercise levels due to their direct relationship with the body tissues’ metabolic rates involved in the gas exchange system. The fractions of inspired $CO_{2}$ ($FiCO_{2}$) and $O_{2}$ ($FiO_{2}$) define environmental gas concentrations, helping to simulate ventilatory stimuli, such as hypoxia and hypercapnia. Atmospheric pressure ($P_{\mathrm{atm}}$) provides information about the environment reference pressure, which helps simulate different altitude conditions. The pressure at the airway opening (*Pao*) is assumed to be equal to $P_{atm}$ for spontaneous breathing conditions, but its value can be varied to simulate different ventilatory therapies. The total time of the muscular contraction (*Tc*) and the period of the muscular contraction-relaxation cycle (*Tim*) are parameters related to muscle contraction times involved in systemic circulatory activity and characterize different physical activity protocols. Other parameters can also be used to characterize the population or the subject whose cardiorespiratory system tries to be simulated, highlighting the anaerobic threshold (*AT*) and basal respiratory tidal volume (*VTn*) for the exercise stimulus.

#### 2.2.2. Experimental Data

The experimental data correspond to those recorded to build and evaluate the cardiorespiratory model. The records comprise signals of $\dot{V}O_{2}$, $\dot{V}CO_{2}$, $\dot{V}E$, TI, VT, BF, HR, ${\mathrm{PAO}}_{2}$ and ${\mathrm{PACO}}_{2}$; systolic (PS), diastolic (PD) and mean (PM) arterial blood pressures; and environment and subjects’ features such as $P_{atm}$, $FiCO_{2}$, $FiO_{2}$, *AT* and *VTn*. The procedures and characteristics are reported in a previous paper [26]

#### 2.2.3. Simulation, Fitting, and Validation

The simulations performed for fitting and validation processes correspond to a healthy human under rest and aerobic exercise. The parameters’ nominal values correspond to those reported in the model-building work [26]. The experimental values of $\dot{V}O_{2}$ and $\dot{V}CO_{2}$ corresponded to the simulation inputs of the model to establish the stimulus levels. This approach has been used to simulate aerobic exercise in physiological models [20,38,44,45]. It is justified because the physiological response under this stimulus is directly related to metabolic rates of $O_{2}$ consumption and ${\mathrm{CO}}_{2}$ production. In addition, the model does not consider workload as a model input. The simulation time was set at 3000 s for each exercise level to guarantee steady-state reach in all variables under study [26]. Steady-state values were calculated as the signal average at the last minute of each simulated stimulus level.

The experimental data were restricted to *AT* because the model is constrained to aerobic exercise. The cardiorespiratory variables evaluated for the selection and fitting procedures correspond to $\dot{V}E$, TI, BF, HR, PS, PD, PM, ${\mathrm{PAO}}_{2}$ and ${\mathrm{PACO}}_{2}$. These variables were selected because they are the available experimental measurements most commonly reported in cardiopulmonary exercise tests. They also provide information about the mechanisms and controllers of the involved systems [41,42,46]. All the subjects’ measurement trends were used to calculate the PE and CF in the parameter selection and fitting procedures. Regarding the validation process, the same variables were evaluated together with VT. Although VT is part of the available records, it was not considered for the selection and fitting processes due to its direct relationship with $\dot{V}E$ and BF ($\mathrm{VT}=\dot{V}E/\mathrm{BF}$), which are output variables of the model’s respiratory controller [26].

Model simulations and data processing were carried out in SIMULINK/MATLAB^®^. The computational characteristics for the simulation were the same as those reported in the model building [26], corresponding to the numerical solver ODE23 with a variable step size lower than 0.01 s. Simulations were run once because all model equations are determinists.

The specific implementation characteristics for each procedure are described below.


**Classification and Selection of Parameters**


Steady-state predictions under different stimulus levels and parameter values were used for subset selection and sensitivity analysis techniques. The stimulus levels corresponding to rest, intermediate exercise, and anaerobic threshold were sequentially evaluated as steps of the same duration to cover the entire range of aerobic exercise. The steady-state model predictions corresponded to each variable’s mean values at the last minute of each simulated stimulus level. Five variations in the parameters resulting from the selection by role procedure were applied. The variations were uniformly distributed in a range of ±5% regarding the nominal values, since a close variation is expected, considering that the experimental data also correspond to healthy adult subjects. Experimental data average measurements of all registered subjects were used for each variable in the three stimulus levels.


**Model Fitting**


This corresponds to the adjustment of the model parameters following the approaches presented in Section 2.1.2. The boundaries for the searching space are defined around the nominal values of the model parameters. The evaluated model predictions corresponded to the steady-state simulation results under variations of $\dot{V}O_{2}$ and $\dot{V}CO_{2}$. These variations consisted of the same three consecutive, equidistant, and incremental steps previously described for classifying and selecting parameters. The experimental data for comparison were also the same, and the CF implemented corresponds to Equation (23).


**Validation Methodology**



*Steady-State Simulation*


PE values obtained from the steady-state model response to the exercise stimulus were used to evaluate the model’s performance (Equation (24)). The model predictions were obtained under simulated variations of $\dot{V}O_{2}$ and $\dot{V}CO_{2}$ as eight incremental, successive, and equidistant steps from rest to the mean experimental value of *AT*. Regarding the experimental data, the model predictions were evaluated by considering the differences concerning each subject.


*Transient Simulation*


The experimental data dynamic responses of the first exercise phase load step were compared with the simulation results under equivalent variations of $\dot{V}O_{2}$ and $\dot{V}CO_{2}$. All variables’ simulation results are shown as proportional changes concerning their initial value to evaluate the change in magnitude due to the stimulus without considering PE. The settling time was calculated as the time elapsed from the stimulus onset to the time for which the model response reached the tolerance band of ±5% of its final value. Systemic arterial pressures were not included here because transient-state experimental values were unavailable.

## 3. Results

### 3.1. Classification and Selection of Parameters

Descriptions of the model parameters are presented in this section in order to elucidate their selection according to the different fitting approaches. The reader can see the model equations in the Appendix A for detail.

#### 3.1.1. Selection by Role

The cardiorespiratory model has 316 parameters (78 of the cardiovascular system, 155 of the cardiovascular controller, 14 of the respiratory mechanics, 13 of the respiratory controller, and 56 of the gas exchange system) distributed according to the proposed classification presented in Table 2.

Only the parameters with the role of covariates, gain, and thresholds were considered for the model fitting. Among gains and thresholds, the maximal abdominal pressure parameter ($P_{abdmax,n}$) was not considered for the subsequent procedure of parameter selection. It should not be modified for the simulation of healthy adult subjects.

#### 3.1.2. Standardization of Simulation Conditions

The parameters used for the standardization of simulation conditions corresponded to $FiO_{2}$, $FiCO_{2}$, $P_{atm}$, basal respiratory tidal volume ($VT_{n}$), anaerobic threshold (AT), total blood volume ($V_{tot}$) and unstressed blood volumes. $FiO_{2}$, $FiCO_{2}$, $P_{atm}$ correspond to the recorded environmental conditions and were assigned by direct equivalence with the reported information. $VT_{n}$, AT, $V_{tot}$, and the unstressed blood volumes are related to the subjects’ specific characteristics and were estimated using equations applied to the experimental data. The results are reported in the model-building work [26].

According to the above, only 216 parameters were selected for the subsequent parameter selection procedures, reducing the number of parameters considered for the fitting procedures by 31.6%.

#### 3.1.3. Parameters Selected for the Base Fitting Approach

Figure 2a shows the first ten parameters for subset selection, and Figure 2b shows the total sensitivity analysis ranking, in descending order from left to right. The parameters selected for the base fitting approach are highlighted in each ranking.

Eight base fitting parameters were selected from the rankings, three from the subset selection technique under consideration of the tolerance range of $1\times 10^{-12}$ (Table 1 and Equation (4)), and five from the total sensitivity analysis corresponding to those whose sensitivity was greater than 46% of the maximum sensitivity found (Section 2.1.1).

Regarding the subset-selection results, it is highlighted that there is a single significant difference between the parameter in the first position and the others presented, evidencing the difficulty of identifying a single optimal solution for more than one parameter using gradient-based techniques [27]. Regarding the total sensitivity analysis, all the selected parameters directly affect the model’s controllers, which are the ones that have the most significant influence on the model’s base behavior, as expected.

This approach evidences the impact of the modifications on selection methods, mainly those related to the error and stimulus-level evaluation. In this sense, the distribution of the parameters regarding the systems and controllers (5 parameters of the respiratory controller, 2 of the cardiovascular system and controller, and 1 of the gas exchange system) is associated with the error contribution of the related variables [26].

After applying the selection criteria, the number of parameters considered for this fitting procedure was reduced by 97.4%. The selected parameters, the corresponding model equations and mechanisms, and the possible variation range information are presented below.

$KR_{lv}$ is a parameter that describes the dependence of the left ventricle resistance ($R_{\mathrm{lv}}$) on the isometric pressure ($P_{\max,\mathrm{lv}}$) in the cardiovascular system [47]. Its nominal value results from scaling data extracted from animal experiments to reflect changes in ventricle volume in human beings. It has been used in different works without presenting variations [15,18,19,22,47,48], even in studies focusing on personalized cardiovascular models [35]. Considering the simulation results under nominal conditions of $R_{\mathrm{lv}}$ and $P_{\max,\mathrm{lv}}$, and applying the rescaling approach of vascular resistance implemented in [35] concerning the mean total blood volume from experimental data, variations close to 6% of the nominal value could be expected.

$MRBCO_{2}$ denotes the metabolic production rate for ${\mathrm{CO}}_{2}$ in the brain tissue [21] and allows us to relate ${\mathrm{PaCO}}_{2}$ with ${\mathrm{PvbCO}}_{2}$, the ${\mathrm{CO}}_{2}$ brain venous blood pressure. Its reported nominal value is 0.0009 L/s STPD and has not been fitted in the different validation works in which the associated model has been used [14,21,22,45]. Works related to other validated models report values for parameters with the same physiological sense ranging from 0.0007 to 0.00104 L/s STPD [8,19,49,50,51]. Therefore, fitted values between −22.2% and 15.6% of the nominal value could be expected.

$T_{0}$ is a cardiovascular controller parameter representing the heart period (HP) in the absence of cardiac innervation [48]. It is the offset term in the equation that relates to the changes in HP induced by sympathetic and parasympathetic stimulation ($\Delta T_{s}$ and $\Delta T_{v}$, respectively). Its reported nominal value results from a fitting process to mimic animal experimental data [47,52]. It has been used without any modification in different publications related to cardiovascular control models [15,18,19,22,47,48]. Considering this value as a proportion of the reported HP at rest (0.833 s), variations close to –7% of the nominal value could be expected whether the median experimental value of HP at rest (0.775 s) is considered a reference.

$P_{max}$ is a parameter of the respiratory controller that denotes the maximum inspiratory pressure [20]. It relates to the inspiratory muscles’ capacity to minimize the work of breathing. In [20], a nominal value of 150 ${\mathrm{cmH}}_{2}O$ is proposed, and variations of around ±66% of this value are evaluated. Subsequent work has used a value of 50 ${\mathrm{cm}H}_{2}O$ on fitting and validating such a model with healthy subjects [53]. Following the above, although the optimization procedure can identify a value near the nominal value, variations higher than 100% could be expected.

$E_{rs}$ is a parameter that denotes the overall elastance of the ventilatory system. It is used in the model’s respiratory controller and lung mechanics to represent the motion equation of the respiratory system. Its nominal value is 21.9 ${\mathrm{cm}H}_{2}O$/l and agrees with that reported in [20,53] for healthy adult subjects. However, other authors report lower values of 10, 8.55, and 8.52 ${\mathrm{cm}H}_{2}O/l$. According to the above, variations of around –66% of the nominal value could be expected [14,21,50,54].

$KpO_{2}$, $KcCO_{2}$, and Kbg are the parameters of the respiratory controller associated with the control of ventilation. $KpO_{2}$ and $KcCO_{2}$ correspond to constant weighting factors related to the peripheral chemoreceptors for $O_{2}$ and the central chemoreceptors for ${\mathrm{CO}}_{2}$ contributions, respectively. Kbg is an offset term that relates to the blood gas dissociation constant for the controller [26]. These parameters correspond to fitted values to mimic the change in alveolar ventilation from rest. They have not been modified in studies in which experimental data on healthy adult subjects are used [14,21,22,55].

#### 3.1.4. Parameters Selected for the Specific Fitting Approach

Figure 3 shows each variable’s relative sensitivity ranking results after removing the parameters selected for the base fitting approach. They are presented in descending order from left to right. The parameters selected for the specific fitting approach are highlighted in the ranking of each variable.

The results highlighted that not all the parameters selected for each variable corresponded to the first ranking positions, which shows the significant influence of the regulation results between the model’s different systems and controllers. This fact was presented mainly for variables not directly related to the controllers (PM, PD, and ${\mathrm{PACO}}_{2}$), evidencing that the parameters of the main regulatory mechanisms have the most significant influence on the model. The parameter selected for ${\mathrm{PACO}}_{2}$, corresponding to ${\mathrm{VLCO}}_{2}$, is not even among the positions presented in the respective ranking, showing the high sensitivity of the variable to the results of other systems and controllers. The number of parameters considered for this fitting procedure was reduced by 97.2%. The selected parameters, the related model equations and mechanisms, and the information reported regarding their possible variation ranges are presented below.

$G_{T,v}$ corresponds to the weighting factor that relates parasympathetic activity with heart rate regulation in the cardiovascular controller [48]. This parameter was initially adjusted to mimic the humans’ cardiac period’s response in [47]. Its nominal value is not related to direct physiological measurement and has not been modified in subsequent applications of the model [15,18,19].

$R_{sa}$ is a parameter of the cardiovascular system that represents systemic arterial hydraulic resistance [48] and relates $P_{\mathrm{sa}}$ to systemic arterial blood flow ($Q_{\mathrm{sa}}$). Its nominal value was initially computed according to experimental cardiac output and cardiovascular pressure measurements [56]. It has been modified in different cardiovascular system versions due to the inclusion of new vascular beds [47]. The latest reported nominal value corresponds to that defined for the case study model [26]. Although it has not been modified in other validation works [15,19,48,57], a rescaling approach was proposed in [35] for personalized fitting based on experimental data on total blood volume. Following the above, using the average experimental value of total blood volume reported in [26] as a reference, a variation close to 5.42% of the nominal value could be expected.

$K_{E,lv}$ describes the left ventricle’s function at the end of the diastole based on the pressure–volume relationship [47]. Its nominal value was initially identified to fit the exponential relationship proposed in [52] to healthy humans’ experimental measurements. According to the experimental measurements reported in [52,58,59], variations of less than −35.7% and greater than 7.1% of the nominal value could be associated with symptoms of cardiac pathologies.

$\Phi_{max}$ is a cardiovascular controller parameter associated with the vasodilation of the peripheral resistance of the active muscles ($R_{\mathrm{amp}}$) during exercise [26]. It corresponds to the upper saturation bound of a sigmoid function used to describe the vasodilator effect independent of tissue hypoxia during exercise [15,60]. The parameter’s nominal value is not related to a direct physiological measure but results from optimization procedures to imitate experimental data.

$\lambda_{1}$ is a parameter related to the breathing pattern optimizer and corresponds to a weighting factor that relates the mechanical work of the inspiratory phase (${\dot{W}}_{I}$) with the average square magnitude of volume acceleration [26]. Its value was fitted in [14,33] to the experimental data of healthy subjects. Considering their results implies possible variations in the nominal value of around −43%.

$V0_{dead}$ is a parameter of the respiratory controller related to the regulation of $\dot{V}E$ [26]. This corresponds to the offset term of the empirical equation used in [21] to calculate the dead space volume as a function of alveolar ventilation. It has not been modified in subsequent validation studies on healthy subjects [14,22].

n is a parameter related to the breathing pattern optimizer, used as a power index of efficiency factors that relate ${\dot{W}}_{I}$ with $P_{\mathrm{musc}}\left(t\right)$ [26]. Following the values fitted in the validation work by [20,33], a variation ranging from −9.17% to 81.7% of the nominal value could be expected.

$C_{1}$ is a parameter related to the dissociation of oxygen in the blood and denotes the maximum concentration of hemoglobin-bound oxygen [50]. Its nominal value was taken from [61] and calculated from predefined pressure, temperature, and the amount of hemoglobin conditions. Although this value has not been modified in subsequent studies of the same blood gas dissociation model, other values are reported for this physiological measure [62], allowing a variation of approximately −3.6% of the nominal value.

$VL_{CO_{2}}$ is a parameter that relates ${\mathrm{PACO}}_{2}$ with the blood concentrations of ${\mathrm{CO}}_{2}$ and $Q_{\mathrm{pp}}$ [26]. It denotes the lungs’ storage volume for ${\mathrm{CO}}_{2}$, and can be understood as a fraction of the functional alveolar volume of the lung [22,50,51,63]. Therefore, using the alveolar volume values reported in [63], a study based on subjects under exercise as a reference, variations around 50% of the nominal value could be expected.

#### 3.1.5. Stimulus-Related Fitting Parameters

For this approach, the selected parameters relate to each of the eight mechanisms reported for the case study model [26]. Only one parameter for each mechanism’s regulatory activity, which was not selected in the previous fitting approaches, was selected. The mechanisms evaluated were: the central command action (I-EP) and the central respiratory neuromuscular drive (NT) on regulation control activities; the central vasodilatory action on active muscles due to central command (I-$R_{\mathrm{amp}}$); the independent description of venous vascular beds from active muscles (V-$R_{\mathrm{amv}}$); the muscle (MP) and the respiratory (RP) pumps; the neural driving ventilation related to metabolism (MRV); and the respiratory control action based on mechanical work of breathing minimizing (minWOB). I-EP, I-$R_{\mathrm{amp}}$, V-$R_{\mathrm{amv}}$, MP and RP are mechanisms that relate I (exercise intensity) to the regulatory activities of the cardiovascular controller and system. Their parameters were defined and optimized from human and animal experimental data [15]. No parameter was selected concerning blood flow signals in the gas exchange system because it is not considered a mechanism that explicitly relates to the exercise stimulus.

The selected parameters, their corresponding mechanisms, and regulatory activity are presented in Table 3. The number of parameters considered for this fitting procedure was reduced by 95.3% with respect to the total after applying the selection criteria.

The related mechanisms and the information reported on the possible variation range of each selected parameter are presented below.

$\gamma_{sh,max}$, $\gamma_{sp,max}$, $\gamma_{sv,max}$ and $\gamma_{v,max}$ are the upper saturation values of sigmoid functions that relate I to sympathetic and vagal efferent activity. In I-$R_{\mathrm{amp}}$, $g_{M}$ is a static gain that relates I to the effect of tissue hypoxia concerning the regulation of the active skeletal muscle’s peripheral resistance ($R_{\mathrm{amp}})$. In V-$R_{\mathrm{amv}}$, $k_{r,am}$ is a constant parameter that characterizes the inversely proportional behavior of the active muscles’ venous vascular beds ($R_{\mathrm{amv}}$) concerning the total volume of blood it contains ($V_{\mathrm{Tamv}}$) during exercise. In MP, $A_{im}$ is a parameter that denotes the peak value of $P_{\mathrm{im}}$, which affects the vascular venous pressure during exercise ($P_{\mathrm{amv}}$). Finally, in RP, $g_{abd}$ and $g_{thor}$ are constant gain factors that relate changes in the tidal volume with the maximum and minimum values of $P_{\mathrm{abd}}$ and $P_{\mathrm{thor}}$, respectively.

$W_{t,sh}$, $W_{t,sp}$, $W_{t,sv}$ and $W_{t,v}$ are weighting factors that relate Nt to sympathetic and parasympathetic efferent activity. Their values were optimized in [50] to ensure that the dynamic behavior of the model under various conditions remains realistic, and they have been used in later works without any modification [14,22].

KcMRV is a parameter of the respiratory controller associated with ventilation. It corresponds to a constant gain that relates MRV to $\dot{V}A$. It was initially defined as equal to 1 in [21] under the consideration of a direct action of MRV. However, in the case study model, it was defined as a constant parameter to consider a weighting factor for the related mechanism.

$\lambda_{2}$ is a factor that weighs the expiratory mechanical work (${\dot{W}}_{E}$) in the total mechanical work of breathing (${\dot{W}}_{T}$). According to [14,20,33], a variation ranging between −28.4% and 170% of its nominal value could be considered.

### 3.2. Model Fitting

A general evaluation range of ±30% regarding the nominal value was defined for those selected parameters whose variation could not be established or constrained to reported values in the literature. For that, the following considerations were taken into account: (a) it is suitable for the constrictions of each associated mechanism of the model, and (b) it agrees with the expected closeness of the results for the nominal values. It was also considered that these parameter values are related to subjects with physiological characteristics similar to the subject’s experimental data (healthy adult males). The evaluation ranges for the other parameters are presented in Table 4, following what was previously described in the selection parameter results.

It should be noted that most of the selected parameters were adapted to the defined general evaluation range; the remaining ones were mainly related to the increase of either lower or upper bounds. Only $K_{E,lv}$ required a decrease in the range due to its relationship with cardiovascular diseases.

Table 5 compares the nominal parameter values against the best optimization results for each fitting stage. Most optimizations were between ±15% of the nominal value, confirming the expected closeness, considering that the nominal values and the experimental data are related to subjects with similar physiological characteristics. The most significant changes that occurred in the second and third fitting stages related to the specific and stimulus-related fitting approaches. The following main modifications were obtained regarding the specific fitting approach: (a) a decreased effect of volume on left ventricular pressure due to the decrease in $K_{E,lv}$; (b) an increase in VD, and therefore an increase in $\dot{V}E$ not related to $\dot{V}A$, due to the increase in $V0_{dead}$; (c) a decrease in the ${\mathrm{PACO}}_{2}$ due to the decrease in $VL_{CO_{2}}$.

For the Stimulus-related fitting approach, the following results were obtained: (a) a decrease of the sympathetic activity related to the regulation of peripheral resistances and heart elastances and an increase for the sympathetic activity for venous volumes due to the modifications in $\gamma_{sh,max}$, $\gamma_{sp,max}$ and $\gamma_{sv,max}$; (b) a decrease in the tissues’ hypoxic effect on the vasodilation action of $R_{\mathrm{amp}}$ due to the decrease in $g_{M}$; (c) an increase in the muscular and respiratory pump activity on venous return due to increased $A_{im}$ and $g_{abd}$; (d) a decreased weighting of ${\dot{W}}_{E}$ regarding ${\dot{W}}_{T}$ due to decreased $\lambda_{2}$. The results obtained from NT are mainly related to increased sympathetic activity regarding heart elastances, but the I-EP results overshadowed this effect.

### 3.3. Validation

#### 3.3.1. Steady-State Response

Figure 4 presents the steady-state model predictions under nominal conditions and at each fitting stage. Eight equidistant step inputs of $\dot{V}CO_{2}$, from 0.3 L/min to 1.0 L/min were used.

The steady-state results confirm the overall improvement of the model predictions after the fitting stages. The base fitting stage improves respiratory variables and gas exchange predictions during rest and exercise. Similar behaviors as nominal results are observed, but with an offset change, wherein the model results are closer to the experimental data’s mean trend. The predictions of cardiovascular variables showed only an improvement in PS. The model’s specific fitting improves most respiratory and cardiovascular predictions, evidencing behavior modifications under the stimulus’s increase concerning the base fitting results.$\dot{V}E$, VT, BF, PS, PM, and PD showed improvements mainly regarding rest and moderate stimulus levels, while HR improved his behavior for high levels of exercise. The stimulus fitting does not significantly improve the model accuracy but primarily benefits the systematic blood pressure predictions.

Figure 5 shows the PE values obtained from the model predictions under nominal conditions and at each fitting stage. The mean, median, interquartile range, overall values, and statistically significant differences among the predictions are presented.

The PE results confirm the steady-state predictions’ observations. The highest errors are obtained under nominal conditions, and the last fitting stage reduces the overall PE by 21.9%. The most significant PE value changes are presented for the predictions of respiratory variables and systemic blood pressure measurements, with statistically significant differences for $\dot{V}E$, VT, PS, PM, and PD. Although the results for ${\mathrm{PACO}}_{2}$ and ${\mathrm{PAO}}_{2}$ do not show significant differences between consecutive fitting stages, a lower dispersion for ${\mathrm{PACO}}_{2}$ and an improvement of 0.07% and 39.30% was obtained for ${\mathrm{PACO}}_{2}$ and ${\mathrm{PAO}}_{2}$ predictions, respectively. The most significant decrease in the overall PE was obtained for the specific fitting approach, which also had the most negligible negative effect on the previous prediction results, followed, in order, by the base fitting approach and the stimulus-related fitting approach.

Table 6 presents the PE mean and standard deviation values for the model predictions under nominal conditions and at each fitting stage for the related subsystems.

In general, a significant decrease in PE was found at each fitting stage. Slight adverse effects were evidenced at the base and stimulus-related fitting approaches. Improvements associated with cardiovascular variables were mainly obtained in the last fitting stage. They were related to the effects of exercise mechanisms on systemic arterial pressures. The most significant results for gas exchange predictions were obtained after the model’s base fitting. The other stages do not present significant variations in PE, as Figure 4 and Figure 5 show. The results related to respiratory mechanics present the most significant variations between stages. This subsystem has the highest contribution to PE under nominal conditions and was reduced by around 30% after the specific fitting approach.

#### 3.3.2. Transient Response

Figure 6 presents the model transient results under nominal conditions and at each fitting stage for a single-step load simulation. Experimental and model-predicted data are depicted as proportional changes to the variable’s initial values. Comparisons for every variable are presented, even though the experimental record length was shorter than the model simulation time.

The simulation results do not present significant variations for the different fitting stages, consistent with the proposed fitting strategies focused on predictions in the steady state. As part of the validation strategy, it is shown that the predictions had consistent behaviors regarding the dynamic response of the experimental data, mainly highlighting the similarity for HR and $\dot{V}E$. The rest of the predictions do not entirely mimic the dynamics of the experimental data.

Table 7 shows the model predictions’ settling times for a single load step simulation under nominal conditions and at each fitting stage. The settlement time results showed changes among the fitting stages. The obtained time values show adverse effects on the respiratory variables’ predictions and significant improvements concerning the variables of the gas exchange system. No significant effect was obtained for HR.

## 4. Discussion

### 4.1. Parameter Classification

This work presents a methodology focused on selecting reduced sets of model parameters that can be reliably fitted in steady-state conditions following several classification approaches. The methodology is based on applying complementary and sequential techniques of parameter classification that evaluate criteria, such as the role in the model, identifiability, and sensitivity to its variation. It involves modifications to reported techniques that evaluate different stimulus levels, imply variations in the parameter values, and consider the experimental data by evaluating the error contribution.

The parameters’ role classification initially constrained those parameters that should be considered for subsequent selection and fitting procedures. Five role classification groups of the parameters most commonly found in mathematical models of physiological systems were proposed. They were based on time constants, conversion parameters, covariates, initial values, and gain and thresholds that facilitated the case study model’s parameter selection.

The parameter identifiability classification was based on Jacobian matrix analysis, which focused on determining which parameters can be reliably fit from available experimental data. Different stimulus levels were evaluated, considering their effects on the model’s prediction behavior. The first three parameters, classified as most identifiable for the case study, were selected (Figure 2a). Each one belongs to one of the main subsystems of the evaluated model (cardiovascular, respiratory, and gas exchange), and their well-conditionality for reliable estimation could be related to the order of magnitude of their nominal values (Table 5). A single significant difference between the parameters in the first position concerning the others presented shows the difficulty of finding a unique identification solution considering more than one parameter (Figure 2a) [27]. This fact is related to the model’s limitation to better mimic the experimental data regarding the results under nominal conditions, which could already be considered sufficiently close. The above can be related to the small decreases in overall PE at each fitting stage (Figure 5) and was involved in the value selection of termination tolerance in the function evaluation for the optimization algorithm (Table 1).

Parameter sensitivity analysis is based on finite difference measurements. It evaluates model variable predictions in steady-state conditions under parameter variations and different stimulus levels. In particular, steady-state conditions allowed considering variable magnitude changes instead of changes to temporary lags among simulated variables, mainly observable in variables with oscillating dynamic behaviors such as $P_{\mathrm{sa}}$, BF, ${\mathrm{PAO}}_{2}$ and ${\mathrm{PACO}}_{2}$. Normalization measures were applied to the obtained relative sensitivities for each stimulus level and variable to obtain unbiased results (Equations (15) and (19)). Finally, given that the sensitivity analysis does not consider the closeness between the predicted and experimental data, PE for each case (variable and stimulus level) was used as a weighting factor to highlight cases where the error was exceptionally high. This consideration results in classifications that are more consistent with the selecting and fitting procedures (Equations (15)–(22)). These modifications are related to obtaining the most respiratory control and mechanics parameters in the first positions of the total sensitivity ranking (Figure 2b), considering that their predictions provide the most significant error under nominal conditions at different stimulus levels (Figure 5 and Table 6). For the specific rankings by variable, obtaining parameters related to the own systems and controllers in the first positions is related to the equitable evaluation of the sensitivities, except for ${\mathrm{PAO}}_{2}$ and ${\mathrm{PACO}}_{2}$ due to their high sensitivity concerning variables from other systems (Figure 3h,i).

### 4.2. Parameter Selection

Parameter selection based on the parameter’s classification according to their role in the model allowed a significant reduction of the complexity associated with subsequent selection and fitting procedures, mainly evidenced by the computational cost involved in the simulations. Its correct application depends on the model mechanisms’ knowledge, its parameters, and the conditions to be simulated because some parameter variations may not be adequate or relevant for the stimulus, environmental conditions, and population features to be represented. Only parameters related to the response in magnitude were selected in this work, which is consistent with the model predictions in steady state and allowed to reduce the PEs in the fitting stages (Figure 4 and Figure 5, and Table 6). In applications focused on fitting the temporal response of model predictions, it is necessary to consider parameters related to time-dependent characteristics.

The procedure results based on the base fitting approach were mainly related to the respiratory system parameters and controller parameters, which is coherent considering that the model predictions under nominal conditions provide the highest PE values (Figure 4 and Figure 5, and Table 6). The model selected parameters agree with the most influential ones, both for its systems and controllers, and their influence on the regulations with the most significant impact on all subsystems in closed-loop: $KpO_{2}$, $KcCO_{2}$ and Kbg of the ventilatory controller, related to $\dot{V}E$; $E_{rs}$ and $P_{max}$ related to BF, VT, and TI; $T_{0}$ related to HR; $KR_{lv}$ related to systemic arterial pressures and blood flows; and $MRBCO_{2}$ related to MRV.

Parameter selection based on the specific fitting approach was proposed to obtain optimal results focused on decreasing the PE values for each variable without negatively affecting the rest of the predictions, as usually happens under the traditional fitting approach. For the case study model predictions, a single parameter per variable was chosen based on its relative sensitivities (Figure 3). Most of the selected parameters agreed with the first positions in the rankings. They belonged to the controllers and were related to the essential regulations of the model in a closed loop. In contrast, the parameters selected for PM, PD, and ${\mathrm{PACO}}_{2}$ were not the most relevant for their rankings because of the high influence of the system controllers in such variables (they can be simultaneously affected by a considerable number of parameters of different subsystems and controllers). Following the above and adding that several variables can belong to the same system or controller, some of the selected parameters were found in relevant positions of other rankings, which could be related to the adverse effects obtained for PM, TI, and ${\mathrm{PACO}}_{2}$ in the second fitting stage (Figure 5 and Table 6).

Parameter optimization under the stimulus-related fitting approach aims to improve the behavior of predictions. This procedure is justified for the case study, considering that although the model was previously evaluated [26], the mechanisms related to the exercise stimulus have not been fitted considering the experimental data. Due to the above, parameters directly related to the effect of the mechanisms on regulatory activities were selected. The fitting of these parameters mainly affected the systemic blood pressure variables (PS, PM, and PD), which agrees with what was expected because most of the mechanisms are related to regulating blood flows and cardiovascular resistances (Figure 4f–h and Figure 5, and Table 6).

### 4.3. Model Fitting

The fitting procedure was proposed as a sequential application of three stages based on different approaches that improved the model’s prediction capacity. The order of the stages was selected considering, firstly, reducing the overall PE, bringing the general trends of simulation results and experimental data closer; secondly, decreasing specific PEs with the least possible adverse effect between predictions; and finally, improving the behavior of the predictions regarding the stimulus. The stages’ results are related to the modification of the values of the selected parameters for each one, considering the constriction of the evaluation ranges for optimization according to the function in the model and the physiological sense (results Section 3.1 Classification and Selection of Parameters and Section 3.2 Model Fitting).

The stage based on the base fitting approach presented an improvement in steady-state respiratory and gas exchange variables. The predictions’ displacements characterized the results without significant changes in the behaviors regarding the stimulus (Figure 4a–e,j). The best respiratory variable results are mainly related to most of the respiratory controller parameters selected for this stage. The decrease in $P_{max}$ promotes the respiratory controller to decrease TI and increase BF [20], which, considering its linear behavior regarding the stimulus (Figure 4c,d), favors a constant increase in $\dot{V}E$ at each level (equations of the ventilation controller in the Appendix A). The increase in $\dot{V}E$ involves more appropriate gas exchange, evidenced by decreases in ${\mathrm{PACO}}_{2}$ and increases in ${\mathrm{PAO}}_{2}$. Feedback from the gas exchange allows regulating $\dot{V}A$ in the respiratory controller, according to the changes in $KcCO_{2}$, $KpO_{2}$, Kbg and $MRBCO_{2}$ (affecting MRV) (equations of the ventilation controller in the Appendix A). Concerning the cardiovascular variables, the overall decrease in systemic blood pressure measurements (PS, PM, and PD) and the non-significant change in HR are mainly due to the decrease in $KR_{lv}$ (heart’s equations in the Appendix A) and the slight increase in $T_{0}$ (effector’s equations for reflex control in the Appendix A), respectively, in an attempt by the model to maintain adequate blood flow values for gas exchange.

The stage based on the specific fitting approach presented improvements in most predictions concerning what was previously obtained (Figure 4 and Figure 5). This is related to the model’s capacity for adapting mechanisms related to each variable by parameter modifications with minimal interference in the other predictions. The adverse effects presented can be attributed to the optimization difficulty concerning two main aspects: similarities in the sensitivity rankings of parameters for different variables and the effect of other predictions on regulation processes. Regarding TI, the parameters selected for $\dot{V}E$ and BF have important positions concerning their sensitivity ranking. They are variables regulated by the same controller (Figure 3e–g), which, in turn, are involved in the regulation of themselves (equations of the respiratory controller in the Appendix A). The ${\mathrm{PACO}}_{2}$ ranking of sensitivities showed that due to the similarity concerning the rankings of the respiratory variables and the low importance of the parameters related to its regulation (Figure 3h), its prediction is mainly affected by other predictions. The improvements in blood pressure predictions are mainly related to the increase in cardiac pressure and the decrease in arterial and peripheral resistance. The increase in $K_{E,lv}$ increases $P_{\max,\mathrm{lv}}$ and compensates for the previous change in $KR_{lv}$ (heart’s equations in the Appendix A), while the increase in $\Phi_{max}$ and the decrease in $R_{sa}$ are related to lower input resistance to the peripheral and arterial beds (equations of blood flow local control and systemic arteries in the Appendix A), obtaining a $P_{\mathrm{sa}}$ signal with an increase in the base level at all stimulus levels (Figure 4g–i). A more significant improvement was obtained regarding HR because the increase in $G_{T,v}$ allows a decrease depending on the parasympathetic activity at each stimulus level (effector’s equations for reflex control in the Appendix A). The correct increase in $\dot{V}E$ is related to increases in $V0_{dead}$ and BF (equations of the ventilation controller in the Appendix A), which, in turn, is related to decreases in TI due to changes in n and $\lambda_{1}$ (equations of the breathing pattern optimizer in the Appendix A). Changes in the behavior of the cardiovascular and respiratory variables regarding exercise were evidenced in this fitting stage and are mainly related to (1) modifying the vasodilation mechanism during exercise due to $\Phi_{max}$ and (2) changing the respiratory controller due to n and $\lambda_{1}$. This promotes similar changes in the gas exchange system, consistent with decreases in ${\mathrm{PACO}}_{2}$ and increases in ${\mathrm{PAO}}_{2}$, also related to increased $C_{1}$ and decreased $VL_{CO_{2}}$, respectively (equations of the gas exchange and mixing in the Appendix A).

The parameter optimization of mechanisms directly related to the stimulus presented improvements mainly for systemic blood pressure measurements because most of them are related to regulations of vascular resistances and blood flows (Figure 4g–i, and Stimulus-related Fitting Parameters in the Results Section 3.1. Classification and Selection of Parameters). Adverse effects of the fitting were evidenced in $\dot{V}E$, VT and HR, and are mainly related to the optimization priority of cardiovascular parameters. The optimizations related to the cardiovascular controller were mainly characterized by a more significant role of NT and a minor role of I in sympathetic and parasympathetic activities regarding the previous fitting stage (Table 5 and the equations of the efferent pathways in the Appendix A). Besides the decrease in the effect of I-$R_{\mathrm{amp}}$ (Table 5 and equations of the blood flow local control in the Appendix A), these results generated smaller increases relative to stimuli for HR (effector’s equations for reflex control in the Appendix A), peripheral vascular resistances, unstressed venous volumes, and ventricular elastances. The cardiovascular controller regulations and the increased effect of V-$R_{\mathrm{amv}}$, MP and RP (Table 5 and the equations of the systemic peripheral and venous circulation, respiratory pump, and muscle pump in the Appendix A) generated a $P_{\mathrm{sa}}$ signal with higher amplitude and base level at low exercise, and smaller increases as a function of the stimulus. Cardiovascular modifications change the behavior of gas exchange system variables, evidenced by slight improvements in ${\mathrm{PACO}}_{2}$ and ${\mathrm{PAO}}_{2}$, but feedback to the respiratory controller implies a variation in the $\dot{V}E$ slope, moving away from the experimental data at low stimulus levels. $\dot{V}E$ is compensated at high levels of stimulus by the decrease in KcMRV and the increase in BF (Table 5 and the equations of the ventilation controller in the Appendix A), and the latter is due to the decrease in TE, which is an expected result during exercise [42], and is achieved by the decrease in $\lambda_{2}$ (equations of breathing pattern optimizer in the Appendix A).

Some results reached the bounds of the proposed evaluation range, evidencing greater differences than expected between the characteristics represented by the nominal values and the experimental data (Table 5). These results were presented for parameters whose value cannot be compared with physiological measurements, and the reported data are mainly related to optimizations focused on mimicking experimental responses. According to the above, they can be considered candidates from a wider evaluation range in future fitting works for the same model.

### 4.4. Validation

#### 4.4.1. Steady-State Response

A decrease in PE for the entire model for each variable and subsystem was obtained throughout the fitting stages (Figure 5 and Table 6), showing that the model’s parameter optimization allows the experimental data to be more adequately predicted in comparison with nominal values. This result was evidenced mainly by changes in the base trend and the behavior of the predictions concerning the stimulus, presented differently depending on the variables and the stimulus level (Figure 4). Although the predictions have behaviors and values that still differ from the experimental mean trend, all the results reasonably approximate a directionally appropriate trend regarding the stimulus change.

The respiratory variables were adequately predicted at rest, but the PEs increased with the exercise level (Figure 4a–d). The simulation results showed linear behaviors regarding the stimulus. $\dot{V}E$ and VT results were similar to expected, although with higher slopes, and BF and TI disagreed. This fact can be related to the limitations of the fitting procedure and the model mechanisms used to represent the experimental behavior, considering that the main parameters related to the stimulus and the respiratory controller’s feedback had been optimized.

Most cardiovascular variables’ predictions presented a behavior opposite respiratory predictions, characterized mainly by lower PE at high stimulus levels. The simulation results in steady state exhibit non-linear behaviors, with similarities regarding the experimental variables related to the number of different regulatory mechanisms. The differences concerning the mean experimental data showed a priority for adjusting PS at all stimulus levels and a proper prediction for HR, PM, and PD only at medium and high exercise levels (Figure 4f,h,i). This is due to the proper optimization of parameters mainly related to stimulus and PS. Even so, the possible fitting of the mechanisms related to the predictions at rest stands out.

The minor decreases in PE were obtained for the gas exchange system and are related to the ${\mathrm{PACO}}_{2}$ and ${\mathrm{PAO}}_{2}$ predictions under nominal conditions already presented the best fit compared to the other variables (Figure 5 and Table 6). Considering that the regulated variables in this system correspond to the most crucial feedback of both controllers of the model, slight variations in behavior and values are significant and related to the improvement of the entire model’s prediction results. Thus, obtaining higher ${\mathrm{PAO}}_{2}$ and lower ${\mathrm{PACO}}_{2}$ values agreed with the expected response of healthy subjects under the simulated stimulus levels and allowed better predictions for the other variables [42,46,64]; but, the increase in the PE of these variables regarding the increase in exercise could be related to the presented fitting limitations, mainly regarding the respiratory variables (Figure 4e,j).

#### 4.4.2. Transient Response

The simulation results in the transient regime did not present significant variations in magnitude change or behavior between the model under nominal conditions and in the fitting stages (Figure 6), probably because the strategy procedures were only focused on steady-state fitting. The most significant variations regarding the model results under nominal conditions were related to the settling time, highlighting an increase in the respiratory variables, no significant changes in HR, and a significant decrease in the gas exchange variables (Table 7). These results are mainly related to changes in the magnitude of the predictions. Although parameters related to the model’s temporal behavior were not optimized, the equations that describe the different dynamic behaviors are affected by the transition’s speeds and initial integration values. Even so, the transient predictions exhibited behaviors that, although they did not fully mimic the complex dynamics of the experimental data, reasonably approximated the general trends and are directionally appropriate considering the change in stimulus.

### 4.5. Strategy Application

In this work, a fitting and validation strategy for physiological models with a large number of parameters and with the consideration of different populations and stimuli was proposed. This strategy was implemented in a case study of a cardiorespiratory model to fit and validate it regarding experimental data from healthy adult subjects on incremental aerobic exercise. According to the results, it can be considered that the fitted model agrees with the validation criteria of physiological models [12,25,65] and predicts with reasonable accuracy and precision the experimental data in both transient and steady-state regimes. This result demonstrates the usefulness of the proposed strategy and the procedures that comprise it.

The purpose of the presented strategy was to provide a systematic methodology for fitting and validation that could be implemented generically in physiological system models, considering predictions under different stimuli, subjects, and experimental conditions. This strategy comprises different procedures and modified techniques that improve the prediction capacity for steady-state predictions. A fitting approach based on the transitory regime’s model predictions could be included as future complementary work. In this case, the procedures for classifying, selecting, and optimizing parameters that characterize temporal dynamic behavior should be considered.

It is important to note that the proposed strategy aims to reduce the optimization problem’s complexity, constraining parameter optimization results to values with physiological justification, and reducing the possibility of obtaining multiple fitting solutions. The benefits of its application will depend on (a) the relationship between the quantity and variety of experimental measures available regarding the number of parameters of the model to be fitted; (b) the model’s capacity to mimic the behaviors of the experimental data; and (c) proper knowledge of the physiological system and the mechanisms of the model. The lack of any of these aspects will constitute a limitation in the application of the strategy, and its consequences will be evident in the poor results of applying the procedures and techniques that compose the approach. A constrained number of experimental variables or a low relationship between them and the model parameters would affect the parameter classification techniques, obtaining low and similar sensitivity values and the different optimization stages, and encouraging multiple identification solutions. The inappropriate choice of a physiological model to represent the experimental data will impact the optimization stages, leading to a lack of precision and accuracy of the results in the validation. Finally, the lack of knowledge about physiological systems and the different mechanisms associated with the model will affect the proper classification, selection, and constraint of parameter values.

Due to the improvement of the prediction capacity and the adequate validation results of the cardiorespiratory model of the case study, both the model and the proposed fitting and validation strategy could be implemented in future works related to diagnostic applications, including athlete evaluation studies, detection of cardiorespiratory diseases, and simulation of the cardiorespiratory response under therapeutic treatments. Furthermore, the application of this strategy regarding experimental data from longitudinal records over time could be considered the basis for developing a dynamic fitting strategy. In this sense, the information obtained from the role, sensitivity, evaluation ranges, and parameter optimization at each instant in time could be used to analyze the temporal evolution of physiological system parameters and predict their future value.

The implementation of the proposed strategy mainly involves the development and execution of software based on data storage and processing. The storage corresponds to the digital recording of the data of physiological variables in databases. At the same time, the processing includes the execution of code and algorithms related to the mathematical model of the physiological system under study and the techniques, stages, and procedures defined by the strategy. The recording of physiological measurements and signals will involve the use of applications that concentrate data from different possible sources, such as personal health devices, laboratory equipment, or medical equipment from clinical environments. The execution of the processing software should be semi-supervised, considering the importance of knowledge of the physiological system and its mathematical model, although in future works, training based on artificial intelligence could be evaluated to fully automate the process. New findings, such as model improvements, new parameter classification techniques, optimization, or validation criteria, can be incorporated into the strategy as long as the order described in each main procedure is considered. The results obtained must be interpreted by medical staff and can support clinical decisions regarding diagnosis and health monitoring.

## 5. Conclusions

This paper presents a strategy for fitting and validating physiological models with a large number of parameters under the consideration of different populations and stimuli. The strategy consists of different procedures and techniques that improve the model’s prediction capacity, considering the complexity reduction of the optimization problem, the results of parameter values with physiological justification, and minimization of the possibility of multiple fitting solutions.

The validation results demonstrate that the case study model is adequate for predicting the cardiorespiratory response of healthy adult subjects under rest and aerobic exercise conditions with reasonable precision and accuracy. This model can constitute an analysis and diagnostic tool related to studies and evaluations of the cardiorespiratory system.

## Figures and Tables

**Figure 1 diagnostics-13-00908-f001:**
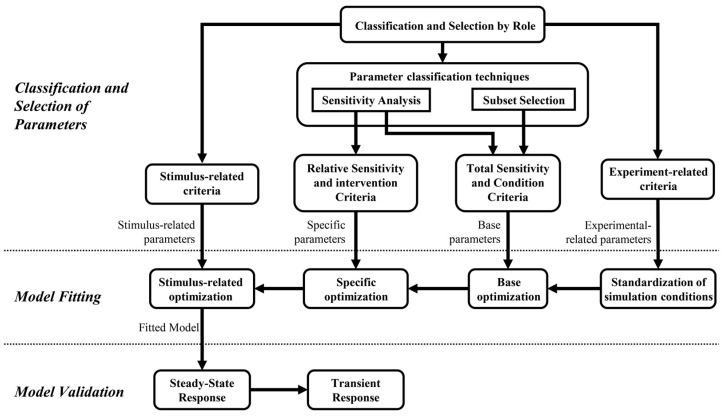
Schematic diagram of the fitting and validation strategy. The dotted lines divide the diagram into the main procedures and their respective stages: the first procedure is at the top and corresponds to the classification and selection of parameters; the second is in the middle and corresponds to model fitting, and the third one is at the bottom and corresponds to model validation.

**Figure 2 diagnostics-13-00908-f002:**
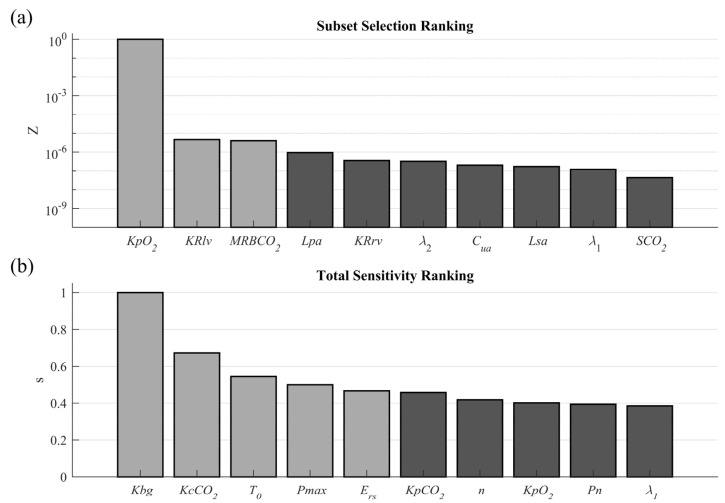
Top ten positions in the rankings of classification techniques: (**a**) correspond to the results for subset selection and (**b**) to the results for sensitivity analysis techniques. The light gray bars correspond to the selected base setting parameters, and the dark gray bars correspond to the parameters that will remain fixed at the nominal values.

**Figure 3 diagnostics-13-00908-f003:**
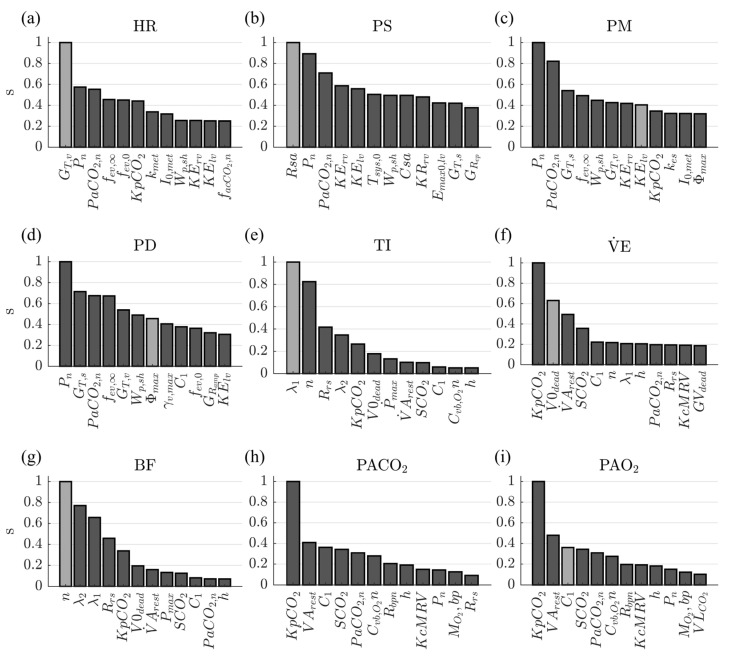
Relative sensitivity rankings after removing the parameters selected for the base fitting approach. The light gray bars correspond to the selected parameters for the specific fitting approach, and the dark gray bars correspond to the parameters that remain fixed at nominal values. The subfigures from (**a**) to (**i**) correspond to the result for each of the variables evaluated.

**Figure 4 diagnostics-13-00908-f004:**
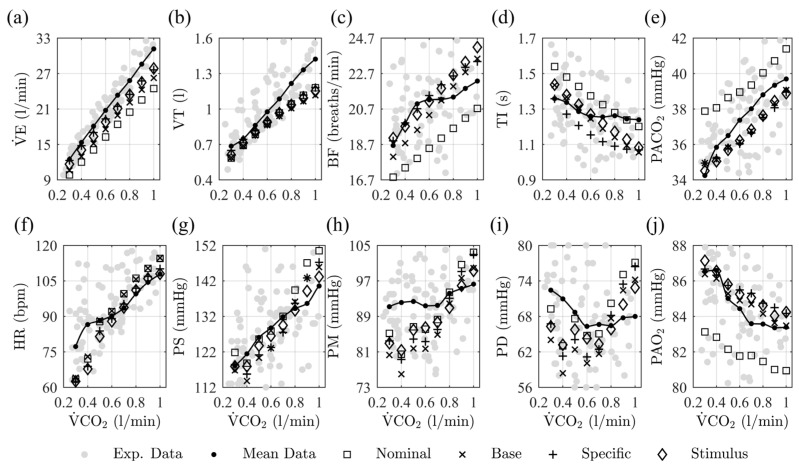
Steady-state model predictions for each cardiorespiratory variable evaluated. Results are shown as a function of $\dot{V}CO_{2}$ values. Gray dots denote the experimental data limited at each subject’s AT; the black dot-line the experimental data average; the square marker the model predictions under nominal conditions (Nominal); the cross marker the predictions of the model at the first fitting stage (Base); the plus sign marker the model predictions at the second fitting stage (Specific); and the diamond marker the model predictions at the third fitting stage (Stimulus). The subfigures from (**a**) to (**j**) correspond to the result for each of the variables evaluated.

**Figure 5 diagnostics-13-00908-f005:**
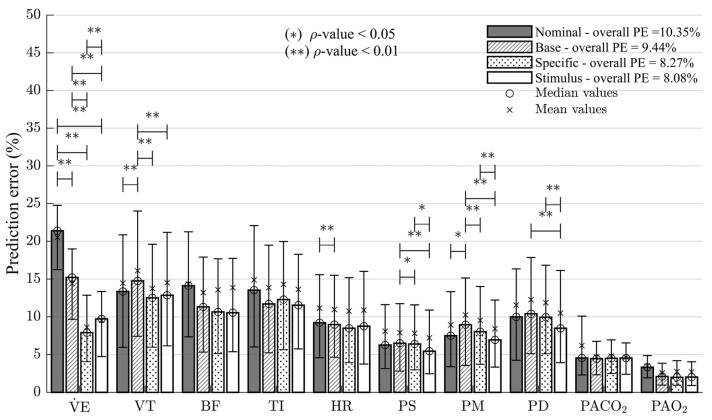
PE results for steady-state predictions under nominal conditions and at each fitting stage. The bar graph represents the errors’ median values, and the whiskers represent the interquartile distance. The gray bars represent the PE under nominal conditions (Nominal), the line pattern bars the results at the first fitting stage (Base), the dots pattern bars the results at the second fitting stage (Specific), and the white bars represent the results at the third fitting stage (Stimulus). Symbols (*) and (**) highlight the statistically significant differences found between the obtained PE values (ρ < 0.05 and ρ < 0.01, respectively).

**Figure 6 diagnostics-13-00908-f006:**
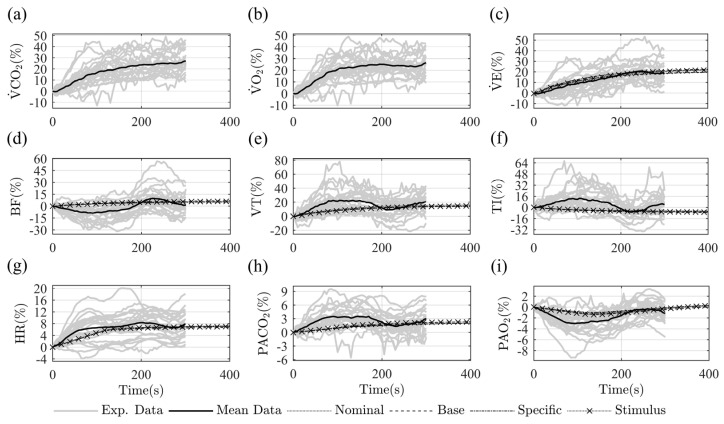
Transient results for a single load step. Model inputs correspond to variations of $\dot{V}O_{2}$ and $\dot{V}CO_{2}$ from 0.64 to 0.82 L/min and 0.58 to 0.76 L/min, respectively. They were obtained from the experimental mean values at the beginning and end of the first exercise load step. The dotted lines are the simulation results under nominal conditions (Nominal); the dashed lines are the simulation results at the first fitting stage (Base); the dash-dot lines are the simulation results at the second fitting stage (Specific); and the dotted lines with cross marker are the simulation results at the third fitting stage (Stimulus). The simulation results are compared with the corresponding experimental data. The solid gray and black lines represent the subject-by-subject experimental data restricted at *AT*, and their total mean value. The subfigures from (**a**) to (**i**) correspond to the result for each of the variables evaluated.

**Table 1 diagnostics-13-00908-t001:** Parameter values used for the CMA-ES algorithm.

Name	Definition	Value
Fitness limit	Value to reach	Infinite
TolFun	Termination tolerance in the function evaluation	$10^{-12}$
TolX	Termination tolerance on x	$10^{-3}$
MaxIter	Maximal number of iterations	$100\times {\;N}^{2}$ *
MaxFunEval	Maximal number of function evaluations	500
MaxRestart	Number of restarts	10
PopSize	Population size	$4+3\ln{\left(N\right)}^{2}$
σ	Initial coordinate-wise standard deviation(s)	0.2(UB−LB) **

* N is defined by the number of the objective variables. ** UB and LB are the upper and lower bounds of objective variables (related to the model parameters’ evaluation ranges).

**Table 2 diagnostics-13-00908-t002:** Distribution of the model parameters according to their role.

Role	Number of Parameters
Covariates	12
Conversion parameters	1
Gain and thresholds	239
Initial values	24
Time constants	40

**Table 3 diagnostics-13-00908-t003:** Selected parameters for stimulus-related fitting.

Mechanism	Parameter	Regulatory Activity
I-EP	$\gamma_{sh,max}$	Sympathetic activity to heart
	$\gamma_{sp,max}$	Sympathetic activity to peripheral resistances
	$\gamma_{sv,max}$	Sympathetic activity to veins volumes
	$\gamma_{v,max}$	Vagal activity
NT	$W_{t,sh}$	Sympathetic activity to heart
	$W_{t,sp}$	Sympathetic activity to peripheral resistances
	$W_{t,sv}$	Sympathetic activity to veins volumes
	$W_{t,v}$	Vagal activity
$I-R_{\mathrm{amp}}$	$g_{M}$	Effect of hypoxia on vascular vasodilation of active muscles during exercise
$V-R_{\mathrm{amv}}$	$k_{r,am}$	Changes in venous resistance of active muscles during exercise
MP	$A_{im}$	Venous return of active muscles
RP	$g_{abd}$	Abdominal pressure signal
	$g_{thor}$	Thoracic pressure signal
MRV	KcMRV	Change in alveolar flow from its resting value
minWOB	$\lambda_{2}$	Total work of breathing

**Table 4 diagnostics-13-00908-t004:** Evaluation ranges of parameters according to the variations and constraints reported.

Parameter	Lower Bound (%)	Upper Bound (%)
$E_{rs}$	−70	30
$K_{E,lv}$	−30	5
$\lambda_{1}$	−50	30
$\lambda_{2}$	−30	200
n	−30	90
$P_{max}$	−30	200
$VL_{CO_{2}}$	−30	50

The values are shown for upper and lower bounds correspond to the percentage of variation regarding the parameter’s nominal value.

**Table 5 diagnostics-13-00908-t005:** Comparison of the parameter nominal values and optimization results for each fitting stage.

Parameter	Nominal Value	Fitted Value	Units
Base fitting approach			
$KpO_{2}$	4.7200 × 10^−9^	4.3473 × 10^−9^	mm Hg^−4.9^
$KR_{lv}$	3.7500 × 10^−4^	3.9120 × 10^−4^	s/mm Hg
$MRBCO_{2}$	9.0000 × 10^−4^	9.3366 × 10^−4^	L/s STPD
Kbg	17.4000	16.6734	Dimensionless
$KcCO_{2}$	0.2332	0.2395	mm Hg^−1^
$T_{0}$	0.5800	0.5877	s
$P_{max}$	50.0000	42.4337	cm H_2_O
$E_{rs}$	21.9000	22.2940	cm H_2_O/L
Specific fitting approach			
$G_{T,v}$	0.0900	0.0951	Dimensionless
$R_{sa}$	0.0600	0.0522	mm Hg·s/mL
$K_{E,lv}$	0.0140	0.0105	mL^−1^
$\Phi_{max}$	20.0000	22.0349	Dimensionless
$\lambda_{1}$	0.8600	0.8901	Dimensionless
$V0_{dead}$	0.1587	0.2059	L
n	1.1010	1.0157	Dimensionless
$C_{1}$	9.0000	9.5133	mmol/L
$VL_{CO_{2}}$	3.0000	2.1479	L
Stimulus-related fitting approach			
$\gamma_{sh,max}$	9.0	6.3	spikes/s
$\gamma_{sp,max}$	5.50	3.85	spikes/s
$\gamma_{sv,max}$	64.90	84.37	spikes/s
$\gamma_{v,max}$	1.9000	2.0708	spikes/s
$W_{t,sh}$	0.4000	0.5043	Dimensionless
$W_{t,sp}$	0.4000	0.4016	Dimensionless
$W_{t,sv}$	0.4000	0.4268	Dimensionless
$W_{t,v}$	0.4000	0.4341	Dimensionless
$g_{M}$	40	28	Dimensionless
$k_{r,am}$	24.1700	27.7488	s/mL
$A_{im}$	50.0000	59.6992	mm Hg
$g_{abd}$	3.3900	4.0826	mm Hg/L
$g_{thor}$	6.800	6.818	mm Hg/L
KcMRV	1.000	0.895	Dimensionless
$\lambda_{2}$	0.4890	0.3423	Dimensionless

Where STPD is standard temperature and pressure, dry.

**Table 6 diagnostics-13-00908-t006:** Prediction error results (%) at each fitting stage for the model subsystems.

Fitting Stage	Cardiovascular	Respiratory Mechanics	Gas Exchange
Nominal	8.33 ± 1.73	15.54 ± 3.87	4.03 ± 0.97
Base	8.71 ± 1.62	13.25 ± 2.02	3.27 ± 1.67
Specific	8.22 ± 1.46	10.83 ± 2.13	3.24 ± 1.78
Stimulus-related	7.41 ± 1.53	11.16 ± 1.35	3.28 ± 1.78

Table 6 shows the model PE under nominal conditions and at each fitting stage. The results are presented in function to the model subsystems: cardiovascular, respiratory mechanics, and gas exchange.

**Table 7 diagnostics-13-00908-t007:** Settling time (seconds) of the model predictions for each fitting stage.

Fitting Stage	$\dot{V}E$	VT	BF	TI	$\mathrm{PAC}O_{2}$	$\mathrm{PA}O_{2}$	HR
Nominal	280.8	277.9	290.4	277.9	2544.6	2831.0	290.6
Base	314.5	314.4	328.2	314.5	2136.1	2795.4	281.9
Specific	295.1	294.0	276.8	266.0	1506.6	2407.5	281.6
Stimulus-related	351.6	351.5	351.6	351.6	1800.6	1622.8	284.3

## Data Availability

The experimental data used in this work correspond to those recorded to build and evaluate the cardiorespiratory model, and the complete description of the procedures and characteristics is reported in a previous paper [26]. The information required to replicate the cardiorespiratory model can be found in the paper of the model building and evaluation [26]. The equations and parameter values to implement and replicate the proposed strategy are described throughout this work and in the reported Appendix A.

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
