# Peer review of "A Novel Strategy to Fit and Validate Physiological Models: A Case Study of a Cardiorespiratory Model for Simulation of Incremental Aerobic Exercise"

_diagnostics, 2023, doi:10.3390/diagnostics13050908_

Round 1

Reviewer 1 Report

The article verifies that there is a high degree of accuracy between the model and the measured data and that it can be used to detect the cardiopulmonary system. The innovation is high, but there are some issues that need to be modified. Minor comments 1. In the summary section, do not use the same connecting words. For example: also. change a synonym. 2. The horizontal coordinates of Figure 2 should be shown horizontally. 3. For each subfigure in the figure, please tabulate the serial number, e.g.: a, b, c ..... Specify this when citing in the text, e.g. Figure 2a. 4. The text removes the see from the brackets. 5.1077-1081 lines, bolded in brackets removed, uniform formatting.

Reviewer 2 Report

In this work, the authors proposed a fitting and validation strategy of physiological models with many parameters, considering different populations and stimuli. The following lists some comments. The first is about the steps of this kind of parameter identification. Although the framework present a strategy, however, the main steps are not clarified. They are not specified as funtional sections, just list all of them. Second, the current parameter identification technique. The comparison study is needed. For different settings, different methods might be useful. Thus, the authors need specify the problems of the proposed method. Third is about the findings. It is good to perform a case study of ardiorespiratory model for simulation of incremental aerobic exercise. However, the implications as well as the medical implications need be strengthened, especailly for the new findings.

Round 2

Reviewer 2 Report

The authors have addressed most of my former comments.